

**Redefining dangerous glacial lakes in Bhutan by integrating hydrodynamic flood**
**mapping and downstream exposure data**
Sonam Rinzin[1], Stuart Dunning[1], Rachel Joanne Carr[1], Simon Allen[2], Sonam Wangchuk[3]
and Ashim Sattar[4]
[1]School of Geography, Politics and Sociology, Newcastle University, Newcastle upon Tyne,
UK
[2]Department of Geography, University of Zurich, Zurich, Switzerland
[3]International Centre for Integrated Mountain Development, Kathmandu, Nepal
[4]School of Earth, Ocean and Climate Sciences, Indian Institute of Technology Bhubaneswar,
Bhubaneswar, Odisha, India
**Correspondence: Sonam Rinzin (s.rinzin2@newcastle.ac.uk)**
**Abstract**
Dangerous glacial lakes in Bhutan have primarily been identified considering the likelihood of
producing a GLOF, which in turn has been assessed only based on upstream lake
area/volume and their surrounding topographic conditions. However, this approach is
incomplete as it ignores the at-risk downstream exposure and vulnerability thus the actual
impacts. Here we redefined dangerous glacial lakes by considering the impact of the simulated
most likely scenario GLOF on downstream exposed elements at risk. Our study shows that a
total of approximately 22399 people, 2613 buildings, 270 km of road, 402 bridges and 20 km$^2$
of farmland are exposed to potential GLOF inundation in Bhutan. We classified lake130
(Thorthormi Tsho) as a very high danger glacial lake in Bhutan, five lakes as high danger and
21 other lakes as moderate danger. Among these high danger glacial lakes, three of them:
lake93 (Phudung Tsho), lake251, and lake278 (Wonney Tsho) were not recognized as
dangerous in previous studies. Our assessment further revealed five downstream local
government administrative units (LGUs) are associated with very high GLOF danger while
nine others are associated with high GLOF danger. Six of these LGUs had not been previously
documented as being at risk from GLOF including: Chhoekhor and Bumthang town in
Bumthang, Paro town and Lamgong in Paro, Nubi in Trongsa and Khoma in Lhuentse districts.
Our study underscores the significance of integrating potential inundation mapping and
downstream exposure data to define dangerous glacial lakes. We recommend strengthening
and expanding the existing GLOF disaster preparedness and risk mitigation efforts in Bhutan
to reduce future damage and loss in high GLOF danger LGUs identified in this study.



## 1. Introduction

There are currently 110,000 glacial lakes globally, with a total area of ~15,000 km$^2$. These glacial lakes have increased in area by ~22% between 1990 and 2020, primarily due to addition of water from the melting of glaciers (Zhang et al., 2024). Glacial lakes across the world have produced 3152 GLOF events between 850 and 2022 C.E (Lützow et al., 2023), which caused more than 12,400 human deaths and damaged infrastructure worth hundreds of millions of USD (Carrivick and Tweed, 2016; Lützow et al., 2023). In HMA, 682 GLOF events occurred between 1833 and 2022, causing 6,907 fatalities (Shrestha et al., 2023) although over 80% of all casualties recorded in HMA are attributed to a single compounding event involving Chorabari Lake in 2013 (Allen et al., 2015; Das et al., 2015). Moraine-dammed GLOF events have caused an order of magnitude greater damage than the combined damage from all other types of glacial lakes, despite moraine-dammed GLOF events only accounting for one-third of total GLOF events in HMA (Shrestha et al., 2023). One main reason is because moraine-dammed lakes are usually located nearer to human settlement than other types of glacial lakes, such as ice-dammed or supraglacial ponds/lakes making it important to quantify the danger it poses to the downstream settlement (Carrivick and Tweed, 2016).

Existing dangerous glacial lakes (DGLs) in Bhutan are defined based on how likely and magnitude of GLOF they will produce, which in turn are assessed based on the inherent stability of the lake's dam and factors that influence the potential for an external triggering event, such as a mass movement entering lake (Allen et al., 2017; Zheng et al., 2021b). Commonly used parameters include topographic potential for mass input into the lake from the surrounding hillslopes, lake volume (usually derived from a relationship to lake area), lake growth, moraine dam geometry and composition, and catchment area (Zhang et al., 2023b; Zheng et al., 2021b). Although the approaches and factors selected are influenced by study objectives and expert judgment, they largely are based on historical events, often backed with limited observed data (Shrestha et al., 2023). Constraining certain parameters, such as the location and magnitude of possible/probable mass movements entering a lake, is challenging even with field-based assessments and more so when using coarse, globally available, open-access data, but the previous study shows that this may be fundamental in the resultant GLOF magnitude (Rinzin et al., 2025). Moreover, the dynamic nature of cryosphere processes, exacerbated under climate warming, means that these reconstructed GLOF characteristics cannot necessarily be applied to contemporary or future conditions (Allen et al., 2017). This is evident from some GLOF events which have occurred from glacial lakes which were deemed less susceptible to GLOF, for example, Lagmale glacial lake in Nepalese Himalaya in 2017 (Byers et al., 2018) and Gongbatongsha Co lake in 2013 in Indian Himalaya (Cook et al., 2018). Thus, the likelihood of producing a GLOF from any glacial lake is subject to inevitable



uncertainties. Most importantly, DGLs defined solely based on GLOF susceptibility of the lake
overlooks how the hydrodynamic properties of a possible GLOF interact with downstream
exposure and vulnerability. If a glacial lake generates an exceptionally large flood, but the
downstream community is unaffected, we can consider the danger from the glacial lake as
low, whereas even a 'small' flood that impacts large number of people should be classified as
high danger. Typically, this is neglected in favour of classifications of danger based only on
lake/trigger conditions, and not downstream impacts (Taylor et al., 2023a).
In recent decades, the amount of infrastructure, buildings and farmland exposed to potential
GLOFs in HMA has increased (Nie et al., 2023). For example, critical infrastructure, such as
hydroelectric power plants are being developed closer to glacial lakes due to growing energy
demand in HMA regions (Nie et al., 2021; Schwanghart et al., 2016). In HMA, the population
in GLOF-exposed areas increased by 0.31% (7.0 million to 9.2 million) between 2000 and
2020 and may therefore have contributed significantly more towards rising GLOF danger than
(debatably) increasing GLOF magnitude due to lake expansion (Taylor et al., 2023b). Thus,
changing downstream exposure and vulnerability can play a greater role in shaping
contemporary and near-future GLOF risk than the glacial lake and surrounding properties,
making the inclusion of the former in the identification of dangerous lakes a crucial, but often
overlooked, factor both in the HMA and other high GLOF risk regions globally, such as the
Andes (Cook et al., 2016; Colavitto et al., 2024).
To identify DGLs with greater confidence and to implement effective management, mitigation
and/or emergency response, we need to consider the interaction between GLOF flow
hydrodynamics, downstream exposure and vulnerability. Taylor et al. (2023a) used
downstream population within a 1 km buffer of the river through which a GLOF would flow, to
a maximum runout of 50 km from each glacial lake to calculate global scale GLOF danger.
However, the coarse resolution of data and crude assumption of GLOF flow path without
hydrodynamic modelling introduces substantial uncertainties due to factors such as detailed
local topography, especially where even populations very close in plain view distance to a
GLOF flow routeway are in reality disconnected from the river by, for example, high river
terraces, which are common in high-mountain region such as Bhutan.  GLOF risk
assessments at the HMA scale have been done by combining hydrodynamic modelling and
open-source downstream data, such as OpenStreetMap (Zhang et al., 2023b). Yet, they
conducted flood mapping only for the glacial lakes that they deemed very high or high danger
through prior GLOF susceptibility assessment. This means that flood mapping for some of the
lakes that can directly impact the downstream communities in case of the future GLOF event
from these lakes have been not carried out despite huge deviation and inconsistencies



between previous susceptibility assessments (Zheng et al., 2021b; Zhang et al., 2023b; Rinzin
et al., 2021; National Centre for Hydrology and Meteorology [NCHM] , 2019). Previous
example of GLOF events from the low GLOF susceptible lake impacting downstream
community underscores uncertainty of these prior GLOF susceptibility assessment (Byers et
al., 2018).  Moreover, since such studies are focused on a global to continental scale, they do
not provide adequate granularity at the national and basin scale for bespoke risk reduction
activities and planning.
This study presents a new GLOF danger assessment approach for Bhutan, which combines
robust flood mapping (through hydrodynamic modelling) and downstream exposure and
vulnerability data. For this, we selected all glacial lakes with an area of 0.05 km$^2$ (n=278) within
the Bhutan Himalaya and conducted hydrodynamic simulation for all these lakes using HEC-
RAS (U.S. Army Corps of Engineers, 2021). We then combined the flood map generated
through hydrodynamic modelling with downstream data on exposure and vulnerability derived
from OpenStreetMap, land use and land cover maps and population and housing 2017 census
data (National Statistics Bureau of Bhutan [NSB], 2018).  As a result, 1) we produced a flood
map for each glacial lake in Bhutan above 0.05 km$^2$; 2) mapped all downstream exposed
elements; and 3) provide a new, updated ranking of glacial lakes in Bhutan, based on the
danger they pose to downstream settlement(s). We have developed a publicly available web
portal that hosts the glacial lake dataset, GLOF flood maps and downstream GLOF risks
across local administrative units in Bhutan.
**2. Study area**
Bhutan's landscape is characterised by high mountains, rugged topography and steep terrain
with elevations ranging between 200 m a.s.l in the south to over 7000 m a.s.l in the north.
Bhutan's northern regions consist of the greater Himalaya mountains, which contain ~1,487
km$^2$ of glacier ice, of which 64% (951 km$^2$) are debris-covered glaciers (Nagai et al., 2016)
(Fig. 1). Between 2000 and 2020, Bhutanese glaciers lost mass at a rate of 0.47 m w.e. yr$^{-1}$,
which exceeds the neighbouring eastern Himalayan (~0.33 m w.e. yr$^{-1}$) and Nyainqêntanglha
(~0.46 m w.e. yr$^{-1}$) regions (Hugonnet et al., 2021). It is projected that Bhutanese glaciers will
undergo continuous and accelerated melting in the future in response to the current climate
warming trend (Rupper et al., 2012).  As of 2020, there were 2,574 (156.63 ± 7.95 km$^2$) glacial
lakes in Bhutan, which was an increase of 17.7% in number and 20.3% in area from the 1960s
(Rinzin et al., 2021) (Fig. 1). While these glacial lakes are predominantly present in basins
such as Phochu (28.18% of the total lake area), and Kurichu (26.35 % of the total area), they
are widespread across the Bhutan Himalaya and drainage from these lakes flow across most
of the major towns and settlements in Bhutan (Fig. 1). Of these glacial lakes, 64 were identified



as highly or very highly susceptible to producing GLOF in the future based on
geomorphological conditions such as topographical potential for avalanching into the lake
(Rinzin et al., 2021).

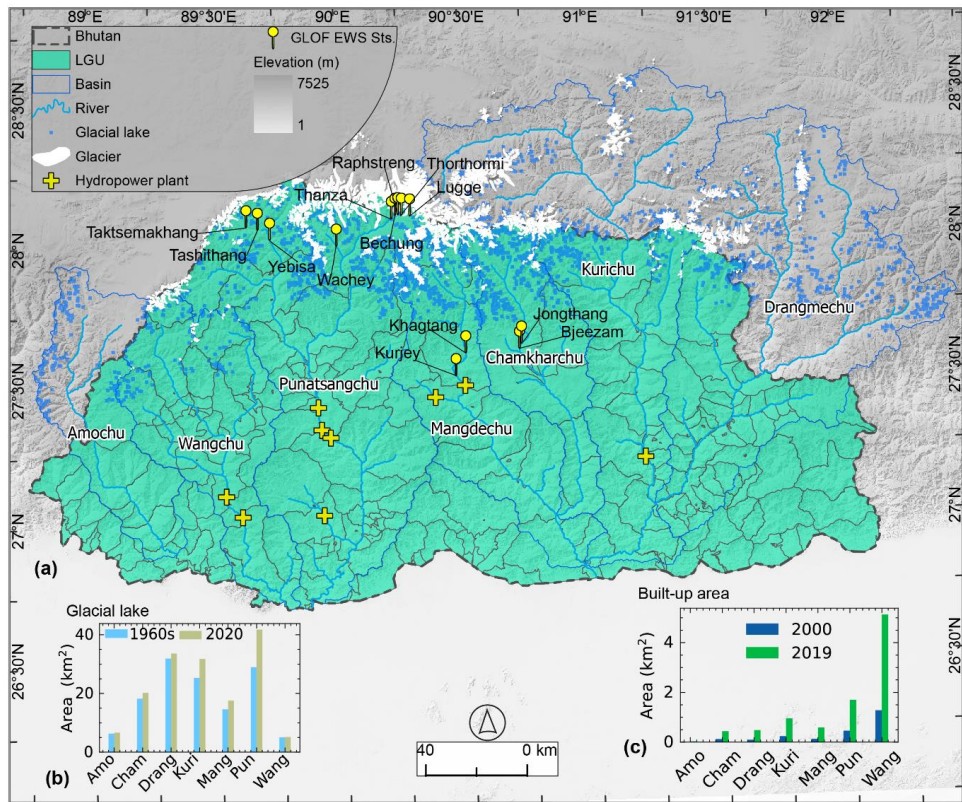

**Figure 1**. The map (a) depicts Bhutan and the glaciated basin from which the river flows into
inland Bhutan. It also shows the distribution of glacial lakes, glaciers, GLOF early warning
monitoring stations (with name of the placed they are located at), and hydropower plants. The
inset bar charts illustrate (b) glacial lake area of the 1960s and 2020 (Rinzin et al., 2021) and
(c) built-up area changes between 200 and 2021 as per the land cover and landuse map of
ICIMOD (Uddin et al., 2021). The basin names are presented in their abbreviated form x-tick
labels for both bar charts.
The glaciers in the northern mountains feed seven major river systems in Bhutan namely
(West-East): Amochu, Wangchu, Punatsangchu, Mangdechu, Chamkharchu, Kurichu and
Drangmechu (Fig. 1). The hydropower generated from these river systems accounts for about
40% of Bhutan's national revenue to a value of 0.27 billion USD (Ministry of Economic Affairs,
2021) as significant power is exported to India. All seven currently operational hydropower





plants and two nearly commissioned hydropower plants (Punatsangchu-I and Punatsangchu-
II) are located along these glacier-fed rivers (Fig.1). The agriculture sector, which is also
heavily dependent on these river systems employs about 60% of the total population (786,385)
(NSB, 2018).  The built-up areas within the 1 km buffer of these glacier-fed rivers have
increased by 200% (2.3 to 9.3 km$^2$) within ~19 years from 2000 to 2019 (Fig.1) (Uddin et al.,
2021). Thus, infrastructure and crucial economic activity have grown rapidly in areas
downstream of glacial lakes in recent decades in Bhutan, making it vital to quantify the risk
posed by GLOFs in these basins.
**3. Datasets and Methods**
**3.1. Lake dataset, drainage volume, peak discharge and flow hydrograph calculation**
We used the glacial lake inventory by Rinzin et al. (2021), which has been developed
specifically for Bhutan, and which offers greater robustness and accuracy for Bhutan
compared to the other datasets available at Pan-HMA scale (Zhang et al., 2023b; Zheng et
al., 2021b) .  Rinzin et al. (2021) mapped 2,574 glacial lakes in 2020, located within 10 km of
glacier termini and with a minimum lake area threshold of 0.003 km². This dataset includes 85
transboundary glacial lakes, located in the Indian and Chinese territories of Himalaya whose
drainage flows into inland regions of Bhutan. Previous records indicate that GLOF originating
from a relatively small lake (as small as 0.001 km²) can cause substantial damage in
downstream communities, although such cases are rare. For instance, across HMA region,
the median area of glacial lakes with known pre-outburst extents is approximately 0.189 km²
(Shrestha et al., 2023). In Bhutan specifically, the smallest glacial lake with a documented
outburst history has a present-day area of 0.0506 km² (Rinzin et al., 2021; Komori et al., 2012).
Including all glacial lakes for detailed hydraulic modelling would substantially increase the
computational demands, whereas resorting to simplified GIS-based approaches (Allen et al.,
2019; Zheng et al., 2021b) to cover all lakes would significantly compromise the robustness
and accuracy of the resulting of flood maps. Moreover, many smaller lakes are unlikely to
generate significant downstream impacts unless they trigger secondary processes (Petrakov
et al., 2020; Cook et al., 2018). Therefore, based on the above empirical evidence, and trade-
off between model complexity and result reliability we focused on glacial lakes that (i) are at
least 0.05 km² in area and (ii) are located within 1 km of glacier termini. This approach ensures
a balance between computational feasibility and the production of reliable flood maps, while
still capturing a substantial number of potentially hazardous lakes. Based on these criteria, we
identified 278 glacial lakes in Bhutan for flood inundation mapping using hydrodynamic
modelling (Fig. S1).



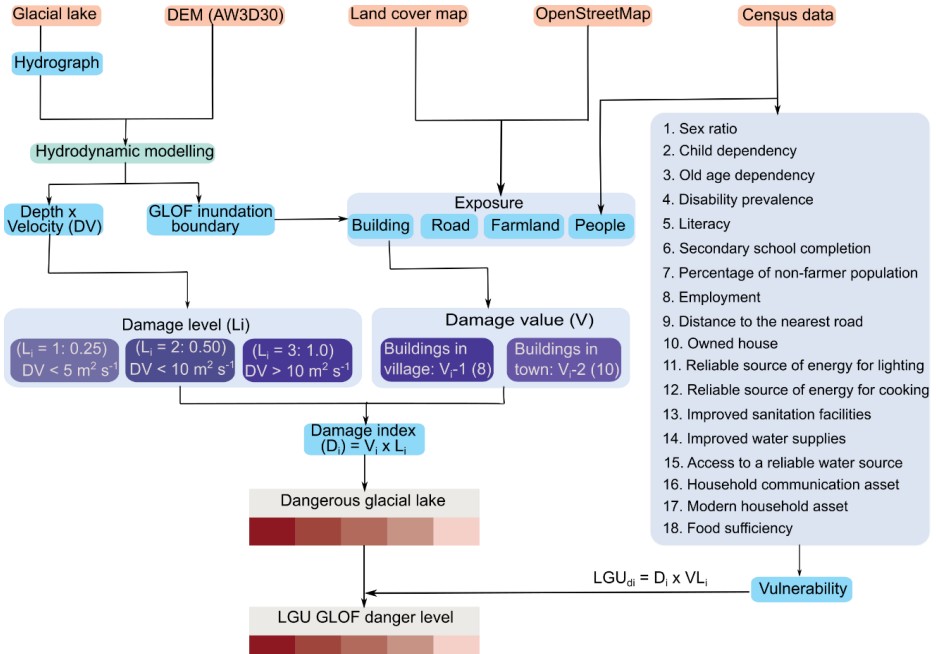


**Figure 2.** Flow chart showing an overview of the methodology we used for assessing GLOF danger in this study. Here GLOF damage index ($D_i$) was calculated as the function of damage value ($V_i$) and damage level ($L_i$). The damage index for the local administrative unit ($LGU_{di}$) is calculated by further multiplying with the vulnerability index ($VL_i$). AW3D30 is abbreviated form for ALOS World 3D - 30m.

$$V = 42.95 \times A^{1.408} \tag{i}$$

Where $V$ is the volume in $10^6$ m$^3$ and $A$ is the area in km$^2$

Accurate glacial lake volume data is crucial as one of the key determinants of modelled GLOF
hydrodynamic characteristics such as flow depth and velocity. However, field based
bathymetric measurement for multiple lakes is costly and not currently possible for some
glacial lakes due to their remote location. In the absence of in-situ bathymetric data to
determine volume, we calculated the volume of each glacial lake using the area-volume
scaling relationship proposed by Zhang et al. (2023a) based on the area of each glacial lake
as mapped in 2020 (Rinzin et al., 2021) (See table S1). This area-volume scaling relationship
(equation (i)), based on recent bathymetric data from the Greater Himalayan region, including
13 representative lakes from Bhutan is well-suited to approximate Bhutanese glacial lake
volumes (Zhang et al., 2023a).



It is important to note that not all lakes drain entirely during a GLOF (Maurer et al., 2020; Nie
et al., 2018; Zhang et al., 2024). However, previous data indicate smaller lakes are more likely
to drain completely during a GLOF event than larger lakes. Here, we used data from Zhang et
al. (2023b) documenting the drainage volumes of 64 lakes in the HMA regions. Among these
64 lakes, the median percentage of drainage volume was 98% for lakes with an area < 0.1
km$^2$, 62% for lakes with an area of 0.1 to 1 km$^2$ and 33% for lakes with an area >1 km$^2$. We
used these observed drainage percentages as the basis to calculate the most likely flood
volume generated by each lake. For simplicity and recognising that these median drainage
values lie within the uncertainty bounds of established area-volume scaling relationships, we
adopted the following assumptions: 100% drainage for lakes < 0.1 km², 60% drainage for lakes
between 0.1 and 1 km², and 30% drainage for lakes > 1 km². Subsequently, we used Evans's
empirical equation (ii) for moraine-dammed lakes to calculate the possible peak discharge of
each lake (Evans, 1986). See supplementary figure 1 (Fig. S1) for the distribution of volume
and peak flow calculated for each glacial lake in Bhutan (see table S1).

$$Q_{max} = 0.72V^{0.53} \tag{ii}$$

$Q_{max}$ is peak discharge, and $V$ is the total volume of the lake calculated using equation (i)

## 3.2. HEC-RAS model set-up

Most GLOFs start from moraine dam breaching, which is frequently triggered by large mass
movement(s) entering the lake from the surrounding terrain hillslopes (Shrestha et al., 2023).
However, conducting a dam breach simulation for each lake is challenging due to complexities
and uncertainties in constraining the appropriate value for a large range of input parameters,
(U.S. Army Corps of Engineers, 2021). To simplify this, we conducted a flood simulation
resulting from each lake by using an input hydrograph as an upstream boundary condition.
For each lake, we generated an input hydrograph by fitting the peak flow of each lake to the
log-normal distribution curve with a standard deviation (sigma) value of 0.75 and a mean of 0,
adapting the approach used by the earlier studies (Carr et al., 2024; Kropáček et al., 2015).
For example, for lake1, the peak flow was 1,110 m$^3$ s$^{-1}$, thus, we constructed a log-normally
distributed hydrograph with a peak flow of 1,110 m$^3$ s$^{-1}$ and gradually decreased the flow after
reaching this peak flow. With this assumption, we generated the hydrograph so that the flow
rises to its peak rapidly and progressively decreases after attaining the peak, which is
consistent with the hydrograph of many previous GLOF events (Maurer et al., 2020; Nie et al.,
2020; Zheng et al., 2021a) (see supplementary figure S2 for representative hydrograph).  The
flow duration of the hydrograph of each lake was subsequently adjusted to account for
the complete drainage of the estimated drainage volume calculated for each lake.  For



example, for lake 1, the required drainage volume was calculated at 1.036 × 10$^6$ m$^3$ and so,
the flow duration for this lake was adjusted so that the cumulative flow through the GLOF event
was equal to this volume (Table S1).
We used the ALOS Global Digital Surface Model (AW3D30) with ~30 m ground resolution as
a source of terrain information for the model setup (Japan Aerospace Exploration Agency,
2021). We chose AW3D30 because various previous studies (Rinzin et al., 2025) have
indicated that it has higher vertical and horizontal accuracy compared to other freely available
DEMs over our study area with similar spatial resolution such as SRTM GL1 (for example, Liu
et al. (2019)). We assigned Manning's n value of 0.06 which is the default value in the HEC-
RAS model set-up (U.S. Army Corps of Engineers, 2021) and has been used in GLOF
modelling in Bhutan previously (Maurer et al., 2020).
We created one HEC-RAS project for each major river basin so that a total of seven project
files correspond to the seven glaciated basins in Bhutan. For each project, the model domain
was established by creating a 1,000 m buffer on either side of the centre line of the river
originating from each lake. Within this model domain, a computational mesh with a grid
resolution equal to the native resolution of AW3D30 (30 × 30 m) was generated. An upstream
boundary condition for each lake was defined at the frontal terminus of each lake. However,
we used the same downstream boundary condition for all lakes in the basin, which were
defined at the furthest end at the international border between Bhutan and India (for example,
Fig. S2). Likewise, unique flow data was created for each lake, where we imposed flow
hydrographs as the upstream boundary condition for the respective lake and downstream
boundary conditions defined by normal depth with an energy slope of 0.01 (U.S. Army Corps
of Engineers, 2021). Finally, one unsteady flow analysis plan for each lake with corresponding
unsteady flow data and boundary conditions was developed. For example, in the Phochu
basin, which contains 67 glacial lakes considered for this study, one project file was
established. This project file included a single model domain, a downstream boundary
condition, 67 upstream boundary conditions and 67 flow data. Accordingly, we created 67
individual plans, each featuring the respective upstream boundary, uniform downstream
boundary condition and flow data that contains the specific hydrograph for each lake (Fig. S2).
We computed all the simulations using the full momentum shallow water equations since it
better represents GLOF rheology than the diffusion wave equation (Sattar et al., 2023; Sattar
et al., 2021).  Following the earlier studies (Rinzin et al., 2023; Maurer et al., 2020), all other
computational parameters were maintained at the default setting. At a mesh size of 30 m, each
model was run stably with a computational time step of 3 seconds within a courant number
well below 2 (U.S. Army Corps of Engineers, 2021). The simulations were executed





simultaneously across 15 computers at the geospatial laboratory in Newcastle University. We
maintained 10 hours of simulation time for each model set-up, which took 2 to 4 hours
depending on the lake's size. Output for each project plan was carefully examined and any
models exhibiting instability (e.g., a courant number above 2 or failed before complete
execution) were re-executed by adjusting the position of upstream boundary condition,
changing the timesteps and adding additional features like refinement regions within the 2D
model domains to ensure stable model run and reliable results (U.S. Army Corps of Engineers,

280    2021).

**3.3. GLOF impact area and exposed elements mapping**
We collated the GLOF inundation boundary for each lake generated through HEC-RAS
modelling and calculated the area and length of each inundation. We calculated the population
density per $km^2$ for each downstream local government administrative unit (LGU) using
population data from the Bhutan 2017 population and census data (NSB, 2018), and from this
population density map, we calculated the number of people exposed located GLOF
inundation extent in each LGU. It is important to acknowledge that the population distribution
data is simplified, although it is the most reliable dataset currently available for Bhutan. We
mapped all buildings, roads, bridges, farmland, and hydropower plants within the GLOF
inundation area to identify downstream elements at risk. We used OpenStreetMap (updated
as of 30-04-2025) to map buildings, roads and bridges. We manually verified the
OpenStreetMap data using Google Earth high-resolution imagery and updated 41 km of
missing roads, 152 buildings and 20 bridges using Google Earth Imagery within the flood
inundation plain. The ICIMOD's Landsat-based land use and landcover map of 2023 was used
to map farmland (Uddin, 2021) since it is of better quality at the HKH scale than other open-
access land cover data, such as Esri Sentinel-2 land cover data (Karra et al., 2021). We
considered buildings the most important downstream exposed element because they are the
primary space where people live mostly. Thus, we used exposed buildings to calculate the
GLOF damage index (Fig. 2).
**3.4. GLOF damage and dangerous glacial lake calculation**
In this study, we defined a dangerous glacial lake based on the downstream damage
(calculated here as damage index ($D_i$)) resulting from each GLOF event (Fig. 2). We assume
that any of our study glacial lakes has the potential to generate GLOF in the future and the
resulting damage will determine how dangerous that glacial lake would be to those
downstream. The $D_i$ for each element (pixel grid) resulting from any GLOF event was
calculated as the function of the value of the exposed element ($V_i$) and the level of damage
($L_i$) following the approach proposed by Petrucci (2012) (equation (ii)) (Fig. 2). Qualitative data



such as construction type, occupancy, and value of the content of the building inside the house
are essential to obtain the appropriate value of each element. However, such qualitative
attributes are incomplete in the existing OpenStreetMap and introduce substantial
uncertainties when estimated employing other open-access data. In this study, our focus is on
providing a relative quantitative comparison of GLOF impacts across different communities
instead of determining exact damage values resulting from each GLOF event. Thus, for
simplicity, we assigned $V_i$ of 8 to the buildings located in the rural areas and 10 to buildings
located in the town areas, following the approaches used by Petrucci (2012) and Carrivick and
Tweed (2016). The categorization of downstream communities into town and rural areas was
achieved by using the local government administrative unit (LGU) map (Fig. 2).

$$D_i \;=\; V_i \times L_i \qquad\qquad\qquad\qquad\qquad\qquad\qquad\qquad\text{(iii)}$$

Where $D_i$ is damage for each downstream exposed element, $V_i$ is the value of each downstream element and $L_i$ is the damage level for each element.

GLOFs with higher water flow velocity can cause more damage to the downstream elements
than slow-flowing water (Federal Emergency Management Agency, 2004). Therefore, we
calculated the $L_i$ associated with each GLOF event as a function of both velocity and depth,
which was accomplished by calculating the depth × velocity ($DV$) from the HEC-RAS output
layer. The Federal Emergency Management Agency (FEMA) of the United States has
established specific depth and velocity thresholds for assessing the collapse potential of
buildings (Federal Emergency Management Agency, 2004). For instance, a one-story wood
building is considered at risk of collapse if subjected to a flood depth of 3 m and a velocity of
1.6 m s$^{-1}$. Applying these thresholds to this study is not appropriate for two key reasons: (1)
the buildings in our study were not classified based on qualitative data such as construction
material or type, and (2) these thresholds may not be directly applicable to the Bhutanese
context, due differences in building design and construction. However, recognizing that higher
$DV$ values correspond to greater damage levels, we categorized the $DV$ values into three
ranges corresponding to three levels of damage: Level 1: 0–5 m² s$^{-1}$ ($L_i$= 0.25), Level 2: 5–10
m² s$^{-1}$ ($L_i$= 0.5), and Level 3: >10 m² s$^{-1}$ ($L_i$= 1) (Fig. 2).
Finally, $D_i$ of all damaged grid cell values within the GLOF path were summed to derive an
overall damage value associated with GLOF from each lake (equation (iv)). This damage value
was then normalized and ranked to classify the relative potential GLOF danger for each lake.

$$G_i = \sum D_i \qquad\qquad\qquad\qquad\qquad\qquad\qquad\qquad\qquad\text{(iv)}$$

$G_i$ is the geographical unit considered here: LGU and dzongkhag. When $D_i$ for a lake was considered, the inundation boundary was considered for $G_i$





**Table 1.** Socio-economic indicators used to calculate LGU's vulnerability to the future GLOF.
The indicators were extracted from Bhutan's 2017 population and housing census. Details on
how data for each indicator are collected are in National Statistics Bureau of Bhutan (2018).
The calculated values were inverted so that they contribute positively to vulnerability for the
indicators other than child dependency, old age dependency, and disability prevalence rate.

| Indicator | Definition |
| --- | --- |
| Sex ratio | Number of males to every 100 females |
| Child dependency | The ratio of the number of children aged 0 to 14 years to population aged 15 to 64 |
| Old age dependency | The ratio of persons 65 years and above to the population aged 15 to 64 years |
| Disability prevalence | The proportion of the population with a disability |
| Literacy | The ratio of the literate population (read and write in Dzongkha and English) aged 6 years and above to the total population of the same age group |
| Secondary school completion | The ratio of persons aged 6 years and above who have completed secondary education (grade XII) to the population of the same age group expressed in percentage |
| Percentage of non-farmer population | Percentage of people aged 15 years who are employed in sectors other than farming |
| Employment | Rate of persons aged 15 years and above who are employed |
| Distance to the nearest road | Households within the 30-minute walk to the nearest road point |
| Owned house | Household living in the owned house |
| Reliable source of energy for lighting | Households with a main source of energy for lighting as electricity |
| Reliable source of energy for cooking | Households with a main source of energy for cooking as electricity |
| Improved sanitation facilities | Households with improved sanitation facilities |
| Improved water supplies | Households with water supplies inside the dwelling |
| Access to a reliable water source | Households with availability of water supplies at least during the critical time (5 AM-8 AM, 11 AM - 2 PM and 5 PM-9 PM) adequate for washing and cooking |
| Household communication asset | Number of communication and media facilities owned by the individual households |
| Modern household asset | Number of modern household assets owned by individual households |
| Food sufficiency | Household having sufficient food to feed all the household members during the last 12 months |

**3.5. Downstream community GLOF damage and danger mapping**
We conducted GLOF damage assessment for downstream settlements at various
geographical scales: 20 districts and 274 local government administrative units (LGUs)
(including 205 gewogs and 69 towns). We aggregated the $D_i$ of each damage grid located
within the respective LGU boundary to calculate the GLOF damage for each LGU using





equation (iv). In cases where downstream elements were affected by GLOFs originating from
multiple lakes, the combined $D_i$ from each contributing lake was considered in the analysis to
account for their exposure to multiple possible GLOF hazards (Fig. 2).
As well as the magnitude of GLOF and the presence of downstream elements along the flow
path, downstream GLOF damage is also determined by the community's capacity to prepare,
respond and recover from a GLOF event (Cutter et al., 2008; Zhou et al., 2009). Lack of this
capacity, often referred to as vulnerability, is influenced by wide-ranging socio-economic
factors including but not limited to the standard of living and gender composition of the
community (Cutter and Finch, 2008). Across the world, developed countries were found to be
more disaster resilient than developing countries while disaster related death and damage
have largely spiked in low-income countries (Rahmani et al., 2022). However, identifying
specific socio-economic variables that are most relevant to GLOF damage remains a
significant challenge, particularly because social data from past events are either unknown or
at times overlooked. Past studies have used variables such as gross domestic product,
population density (Carrivick and Tweed, 2016), human development index, corruption index
and social vulnerability index at the national scale (Taylor et al., 2023a). While such data
represent a broad overview of the country's socio-economic condition and thus vulnerability
to disaster, they do not represent the regional and community level disparity within the country
that influences their ability to respond and recover from the disaster. To address this, drawing
upon our local understanding and following earlier studies (Allen et al., 2016; Rinzin et al.,
2023), we calculated the relative vulnerability index ($VL_i$) using a total of 18 socio-economic
indicators from the 2017 Bhutan population and housing census NSB, 2018) (Table 1). This
census data which is updated every after 10 years represents the most comprehensive and
detailed dataset currently available, offering spatial granularity at the individual LGU level.
These indicators are essential for evaluating a community's preparedness and response
capacity to disaster from hazards like GLOF (Cutter and Finch, 2008). For example, Bhutan's
traditionally gendered societal structure, men often assume more prominent roles in disaster
response efforts. The $VL_i$ for each LGU was calculated as the normalised value across these
18 socio-economic indicators (Fig. 2). The definition and approach used for calculating each
indicator is summarized in Table 1.
Assuming that the LGUs with higher $VL_i$ are the least capable to respond to and recover from
a future GLOF, the damage index ($D_i$) of GLOF for each LGU unit was multiplied with to the
$VL_i$ using equation (v).

$$LGU_{di} = D_i \times VL_i \qquad\qquad\qquad\text{(v)}$$



Where *LGU$_{di}$* is the GLOF damage index for the individual LGUs normalized to their vulnerability index (*VL$_i$*)

### 3.6. GLOF arrival time and GLOF monitoring assessment

Building damage from GLOF is a function of hydrodynamic factors such as depth and velocity. On the other hand, human casualties and injuries also depend on the warning / response time, making it essential to consider flood arrival time in GLOF danger assessment. Thus, GLOF arrival time needs to be considered separately from the *D$_i$* we computed earlier. Accordingly, we determined the flow arrival time of the earliest arriving GLOF for each LGU to quantify the worst-case scenario for LGU exposed to multiple GLOF sourcing lakes.

The National Centre for Hydrology and Meteorology (NCHM), Bhutan monitor several lakes in Bhutan identified as dangerous (NCHM, 2019). Currently they have GLOF early warning system covering, Punatsangchu, Mangdechu and Chamkharchu basins which consists of 23 monitoring stations (Fig. 1). We utilized monitoring station location data from NCHM to evaluate the relationship between Bhutan's existing early warning system, modelled GLOF scenarios originating from these glacial lakes and affected downstream communities. To achieve this, we first located all monitoring stations in Bhutan and counted how many of our catalogues of possible GLOFs in the region are covered by the existing early warning system based on their hydrological relationship (National Centre for Hydrology and Meteorology, 2021). We assumed that if a GLOF flow intersects any of the existing EWS monitoring station, then the event considered to be monitored by the existing EWS in Bhutan. Similarly, if a LGU is affected by a GLOF that passes through one or more of these stations, the associated GLOF danger that LGU is regarded as being monitored by the existing EWS.

**Table 2.** GLOF exposed elements: people, buildings, roads, bridges and farmland distributed across top 20 GLOF LGUs. The total value at end last row represents the total exposed elements across all LGUs in Bhutan.

| Gewog/town | District name | Building (count) | People count | Road (km) | Farmland (km²) | Bridge (count) | Danger (rank) |
|---|---|---|---|---|---|---|---|
| Chhoekhor | Bumthang | 191 | 297 | 41.7 | 0.28 | 36 | 1 |
| Bumthang Town | Bumthang | 283 | 5740 | 13.3 | 2.32 | 2 | 2 |
| Punakha Town | Punakha | 272 | 4911 | 10.8 | 1.14 | 4 | 3 |
| Lunana | Gasa | 121 | 86 | 40.0 | 0.00 | 30 | 4 |
| Toedwang | Punakha | 60 | 86 | 5.0 | 1.08 | 8 | 5 |
| Nubi | Trongsa | 29 | 229 | 2.0 | 0.30 | 8 | 6 |
| Thedtsho | Wangdue Phodrang | 165 | 535 | 3.6 | 0.90 | 6 | 7 |



| | | | | | | | |
|---|---|---|---|---|---|---|---|
| Dzomi | Punakha | 53 | 560 | 6.2 | 1.22 | 6 | 8 |
| Paro Town | Paro | 262 | 5535 | 13.2 | 1.14 | 12 | 9 |
| Wangdue Phodrang Town | Wangdue Phodrang | 109 | 1365 | 1.2 | 0.25 | 0 | 10 |
| Lamgong | Paro | 195 | 303 | 7.9 | 0.85 | 6 | 11 |
| Lingmukha | Punakha | 65 | 41 | 5.7 | 0.19 | 2 | 12 |
| Khoma | Lhuentse | 42 | 49 | 6.7 | 0.04 | 8 | 13 |
| Khatoed | Gasa | 17 | 3 | 0.4 | 0.00 | 4 | 14 |
| Athang | Wangdue Phodrang | 38 | 3 | 2.0 | 0.42 | 10 | 15 |
| Darkar | Wangdue Phodrang | 85 | 21 | 3.2 | 0.31 | 16 | 16 |
| Yalang | Yangtse | 9 | 80 | 0.9 | 0.46 | 6 | 17 |
| Langthil | Trongsa | 11 | 44 | 1.3 | 0.82 | 8 | 18 |
| Saephu | Wangdue Phodrang | 6 | 162 | 14.2 | 0.00 | 4 | 19 |
| Sharpa | Paro | 120 | 168 | 3.8 | 0.94 | 8 | 20 |
| **Total** | | **2613** | **22399** | **265** | **19** | **364** | |

## 4. Results

### 4.1. GLOF impact and exposure

Our study revealed that GLOFs from individual glacial lakes can travel as far as 167 km downstream and can inundate a maximum area of 30 km$^2$. The modelled GLOFs exhibit a median travel distance of 40 km and an inundation area of 2.9 km$^2$ (Fig. S3). Collectively about 2% (781 km$^2$) of Bhutan's total land area is exposed to GLOF. The mean flow depth and velocity were 3.3 m and 3.4 m s$^{-1}$, respectively (Fig. S3). The shortest arrival time to the nearest building was 8 minutes and the longest was 10 hours (Fig. S3). As a result, a total of 22399 people, 2613 buildings, 270 km of road, 402 bridges, 19 km$^2$ of farmland and 4 hydropower dams are exposed to GLOFs in Bhutan (Table 2). Of the total modelled GLOF events, 71% (n = 197) affect roads, 42% (n=116) affect buildings and 28% (n=77) affect farmland. The rest of the GLOFs do not affect any downstream entities. Focusing on Bhutan's most dangerous lake, Thorthormi Tsho, can impact 1119 buildings, 72 km of roads, and 4.2 km$^2$ of farmland making it the most consequential event for reaching elements at risk. It is followed by lake278, located in the Wangchu basin and Chubdha Tsho in the Chamkharchu basin, both which are classified as high danger glacial lakes by this study (Table S1).

Out of 278 glacial lakes selected for flood mapping for this study, 85 (30.6%) are within catchments that cross the boundaries of India and China and drains into Bhutan inland. GLOF from these transboundary lakes also affect substantial number of downstream elements



located in Bhutan, including 20 buildings, 0.6 km$^2$ of farmland, and 2 km of roads in Bhutan.
All these exposed elements are situated within the Kurichu and Drangmechu basins.
The exposed elements are distributed across 17 districts and 88 local government
administrative units (LGUs). Bumthang, Paro, Punakha, Wangdue Phodrang and Gasa
districts are most affected by the GLOFs. For example, Paro itself has 673 GLOF exposed
buildings, 32 km of road and 64 bridges (Table 2). Among the LGUs, the maximum exposed
building is in Bumthang town (n= 283) followed by Punakha town (n= 272) and Paro town
(n=262). The greatest road (roads, footpaths, tracks) inundation occurs in Chhoekhor followed
by Lunana and Saephu, while most farmland is impacted in LGUs such as Bumthang town
(2.3 km$^2$), Dzomi Gewog (1.2 km$^2$) and Paro town (1.1 km$^2$) (Table S2).

**4.2. Dangerous glacial lakes**

We defined dangerous glacial lakes in Bhutan based on the damage index ($D_i$) calculated by
combining flow depth, velocity and downstream exposure data. Of the total lakes studied here
(278), 164 had zero damage index as the GLOF from these lakes does not impact any
buildings. The computed DI for rest of the glacial lakes range between 2 and 3435. Here we
normalized DI value between 0 and 1. With the highest $D_i$, Thorthormi Tsho in the
Punatsangchu basin emerged as the most dangerous high danger lake in Bhutan (Fig. 3).
Additionally, based on DI, we categorized glacial lakes into four danger levels: very high
danger, high danger, moderate danger, and low danger using the Natural Jenks classification
system in ArcGIS. Using this approach, five other lakes are identified as high danger which
are distributed across the Wangchu (2), Chamkharchu (2), and Punatsangchu (1) basins.
Twenty-one of the glacial lakes were in the moderate danger category: four each in
Punatsangchu, Chamkharchu and Drangmechu basins, and six in Mangdechu, two in Kurichu
and one in Drangmechu basins (Fig. 3). The remaining (251) were classified as low danger.
None of the high or very high danger glacial lakes were located within the Chinese and Indian
sides of the basins, which drain into Bhutan (Fig. 3).



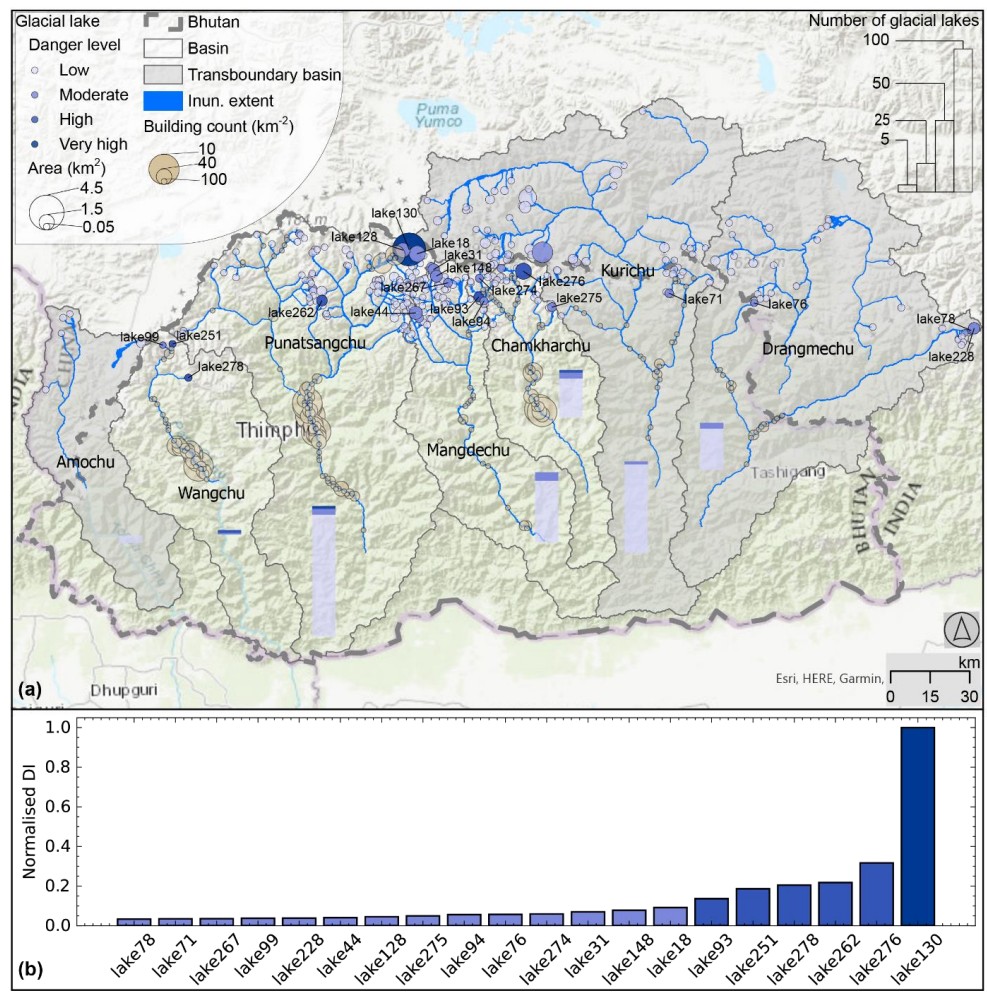

**Figure 3. Distribution of dangerous glacial lake**. The (a) map shows the distribution of glacial lake with associated GLOF danger level across the eight glaciated basins in Bhutan: very high (VH), high (H), moderate (M) and low dangers. The bar charts within the map show the number of glacial lakes (where height of the bar corresponds to number of lakes in each basin referenced to inset bar chart) with various danger levels in each basin. The bubble along the flood path shows number of buildings per km². The (b) bar chart shows the damage index (normalized between 0 to 1) associated with top 20 lakes arranged in the ascending order (left to right). The lake ID on the x-tick labels correspond to the ID on the map. Base map image is the intellectual property of Esri and is used herein under license. Copyright © 2025 Esri and its licensors. All rights reserved.



**4.3. Downstream GLOF danger**

In this section, we present the GLOF damage ranking and level associated with downstream communities, encompassing 20 districts and 274 LGUs. Based on the damage index for respective LGUs (*LGU_di*), which accounts for both damage from GLOF and people's vulnerability (Fig. S4), Punakha is identified as the district that would suffer from the highest GLOF damage in the future followed by Bumthang and Wangdue Phodrang districts.

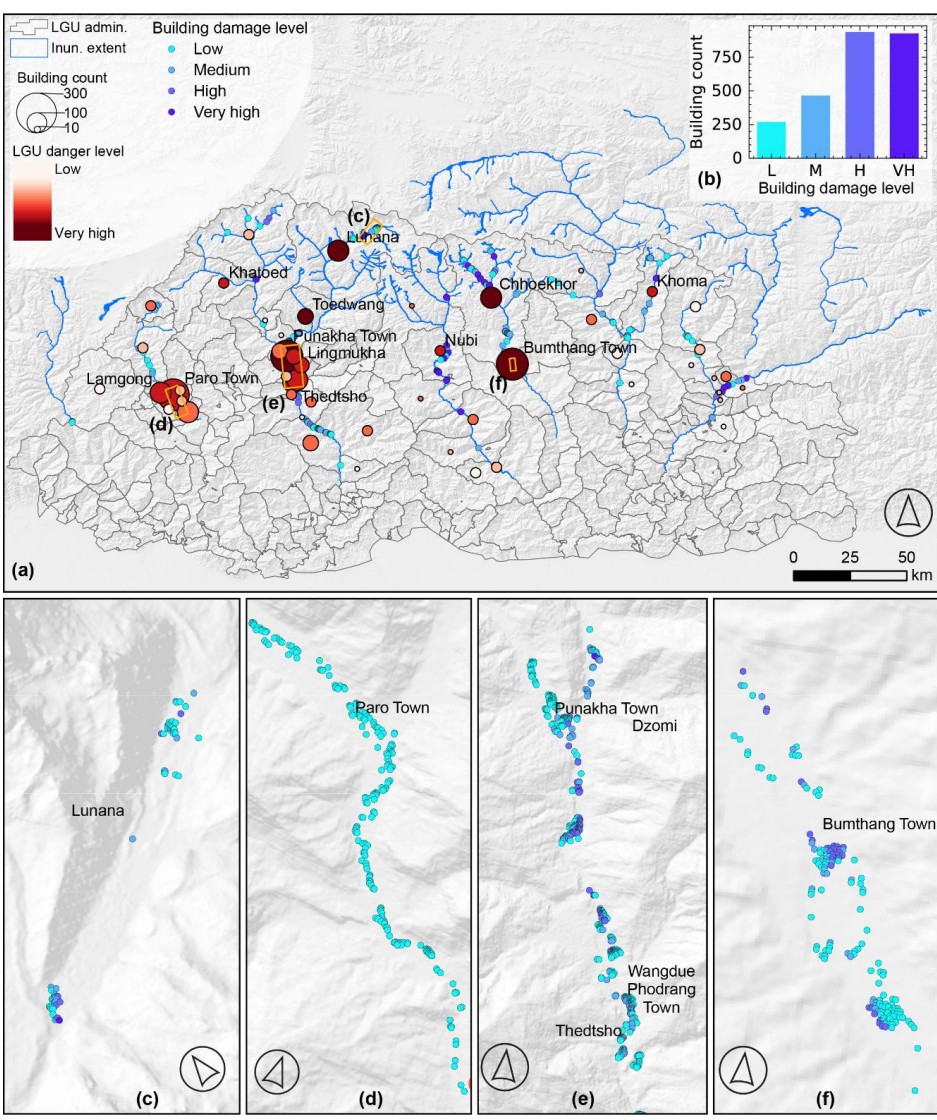

**Figure 4.** Downstream GLOF danger. GLOF danger level across (a) local administration units and GLOF inundation extent (inun. extent). Inset (b) bar graph shows the number of all



buildings in Bhutan impacted by the modelled GLOF and associated damage level: very high
(VH), high (H), moderate (M) and low (L).  The lower panels (c–f) are the zoomed-in map from
panel A which shows the damage level associated with individual buildings.
Among the LGUs, Chhoekhor gewog is associated with the highest GLOF damage followed
by Bumthang town, Punakha town and Lunana gewog (Fig. 4). The classification of $LGU_{di}$
yielded five LGUs associated with very high GLOF damage while nine others were associated
with high GLOF damage. Likewise, 13 LGUs were associated with moderate GLOF damage
while the rest were identified as low GLOF damage LGUs (Fig. 4).
**4.4. Flow arrival time**
We also ranked the LGUs based on the flow arrival time of the GLOF scenario that would
impact the first buildings in the LGU. Results showed that, for the seven gewogs including
Soe, Khoma, Chhoekhor, Lunana, Laya, Saephu, and Kurtoed, the fastest GLOF can impact
some of their buildings within 30 minutes. Some buildings in Khatoed, Tsento, Toedwang and
Nubi could be affected within one hour. Nine gewogs can be affected within 2 to 4 hours,
another nine within 4-6 hours while the fastest GLOF could take more than 6 hours to affect
buildings in other LGUs (Fig. 5).
In the LGUs such as Soe and Laya, the first buildings affected by the GLOF are typically
isolated and located very close to glacial lakes. Despite their proximity, the GLOF danger
ranking for these LGUs remains relatively low due to the limited number of exposed buildings
in these LGUs (Fig. 5). Therefore, we compared the $Di$ and the arrival time of the fastest-
arriving GLOF within each LGU. This analysis identified Lunana and Chhoekhor as LGUs with
very high GLOF danger levels, and the fastest GLOF can arrive in as little as 15 minutes. On
other hand, the fastest GLOF impacting buildings take up to three hours for other LGUs with
very high GLOF danger such as Punakha town and Bumthang town (Fig. 5).
**4.5 Early Warning System and GLOF**
We analyzed the distribution of the existing GLOF early warning system in Bhutan with respect
to dangerous glacial lakes, and downstream communities associated with GLOF danger.
Currently Bhutan has GLOF early warning system in three basins: Punatsangchu,
Mangdechu, and Chamkharchu. Across these basins, the system is equipped with 13
monitoring stations placed at various locations (Fig.1). Assuming that GLOFs from a lake may
be monitored if an EWS monitoring stations is located downstream of the glacial lake, our
study shows existing EWS currently tracks 51 out of the 278 glacial lakes we investigated
here. Among these monitored lakes, the network includes the most dangerous glacial lake,
Thorthormi Tsho, as well as two of the five high danger glacial lakes and five of the six



moderate danger glacial lakes. The remaining monitored lakes are classified as low or very
low danger. Notably the high danger glacial lakes identified here including lake251, lake262
and lake278 are not monitored by the existing early warning systems.

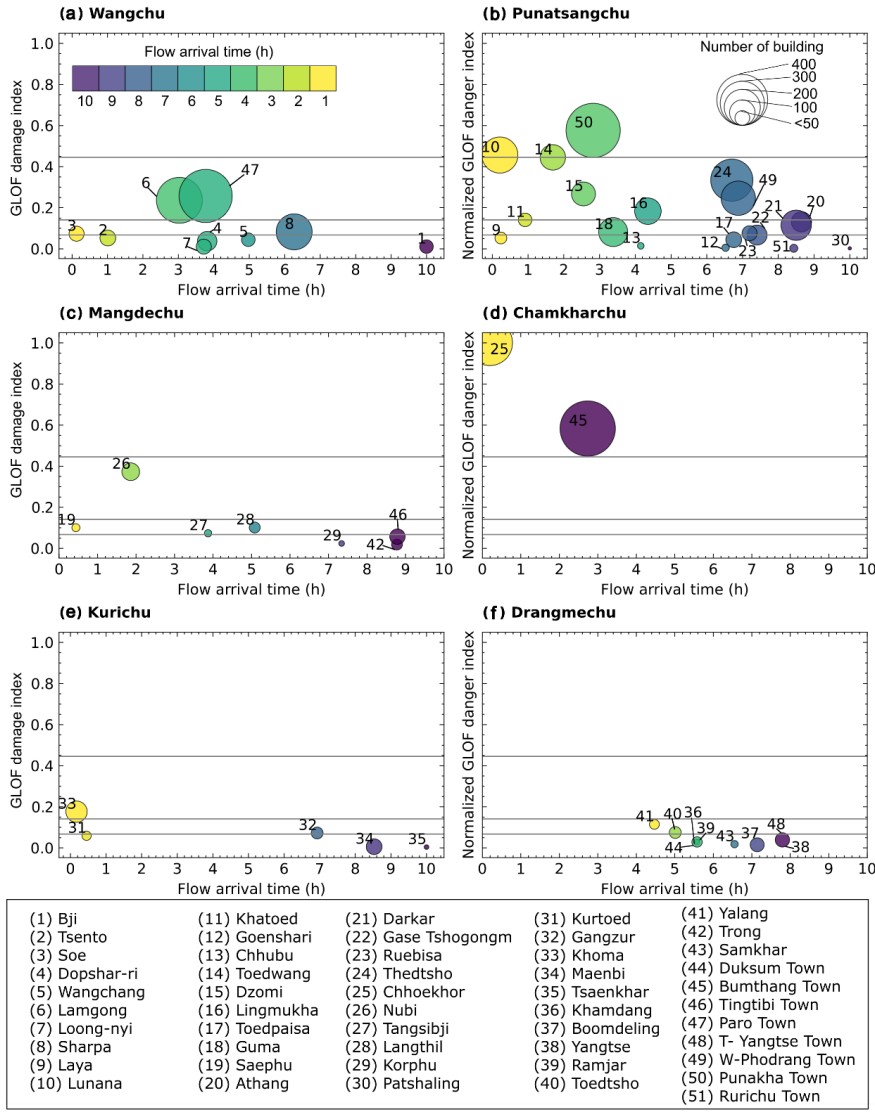


**Figure 5.** The damage index and flow arrival time (of the fastest arriving) GLOF for each local
administrative unit (LGUs) in impacted basins: (a) Wangchu, (b) Punatsangchu, (c)
Mangdechu, (d) Chamkharchu, (e) Kurichu and (f) Drangmechu. Numbers associated with
each bubble in plot correspond to the numbering corresponding to the name of each LGU in
the lower panel. The flow arrival time colour code legend in panel (a) and number of building
legend in panel (b) applies to all the panels. The horizontal grey lines categorize the local
administrative unit (LGUs) into various GLOF danger level based on damage index (low
danger level to very high danger level).

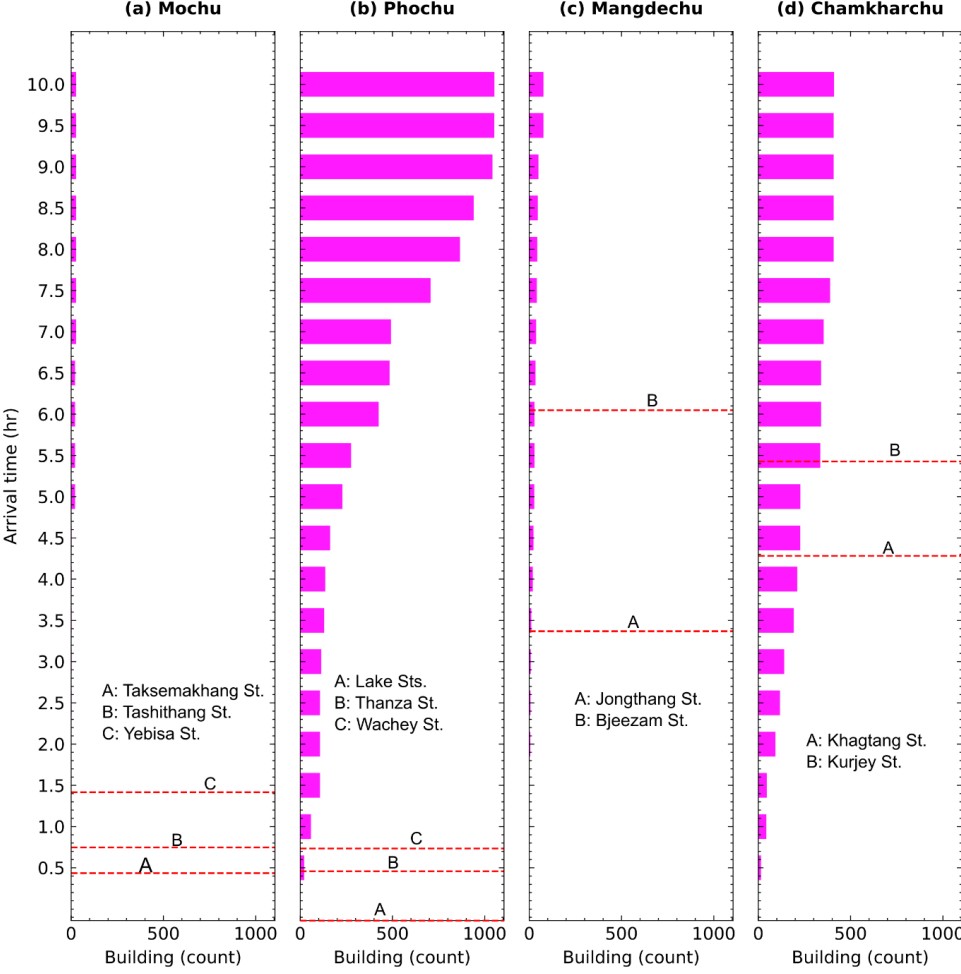


**Figure 6.** Bar plots showing the number of buildings located downstream of GLOF early
warning monitoring stations in Punatsangchu basin [(a) Mochu, (b) Phochu], (c) Mangdechu,
and (d) Chamkharchu basins. Red dashed lines represent the location of EWS monitoring
stations relative to average flow arrival time of all GLOFs detected by each EWS monitoring
station. The bars represent cumulative numbers of GLOF exposed buildings located at the
respective basin where early warning stations are operational. The name location of each
EWS monitoring station is indicated with alphabet (A to C) within the respective panel. The





monitoring station located at the lakes including Bechung, Raphstreng, Thorthormi and Lugge
Tsho in Phochu basin are marked as Lake Sts. in panel (b).
We further examined residents of how many GLOF exposed buildings can receive early
warnings based on their hydrological relationship to the existing EWS monitoring stations.
Assuming that the buildings located downstream of the EWS monitoring stations can receive
early warning, our study revealed that the existing GLOF monitoring stations can provide early
warning to the people living in1549 buildings of which about 75% are in the Punatsangchu
basin. Of these, residents in 268 buildings are estimated to have less than 30 minutes to
evacuate after receiving warning from the EWS monitoring stations located in their respective
communities (Fig. 6).
Conversely, people living in 1050 exposed buildings, that is at least 41% of them do not have
access to early warning coverage. Approximately half of these unserved buildings clustered
in downstream LGUs with high GLOF danger including Lamgong and Paro Town in Paro
districts. Although EWS in place in the Chamkharchu basin, a cluster of about 82 buildings in
Chhoekhor in Bumthang are not covered by EWS. These is because the flood waves from the
potential GLOF can arrive these buildings before activating the monitoring stations at
Khagtang and Kurjey (Fig. 6).
**5. Discussion**
**5.1. Redefined dangerous glacial lake**
Some of the most devastating historical GLOF events in the world have occurred from
seemingly inconspicuous glacial lakes (Allen et al., 2015; Petrakov et al., 2020), whilst some
large-magnitude GLOF events have caused minimal or no downstream damage (Shrestha et
al., 2023; Lützow et al., 2023). This is because the GLOF magnitude alone does not determine
downstream damage caused by the GLOF event, instead, it is the interaction between GLOF
magnitude and the downstream exposed elements that determine the extent of damage
(Taylor et al., 2023a). For example, the greatest structural damage associated with 2023 South
Lhoknak Lake GLOF event in the Indian state of Sikkim occurred between 200 and 385 km
downstream of the glacial lake, with 59% of these impacted structures constructed within the
past decade (Sattar et al., 2025). This highlights the escalating risks posed by infrastructure
expansion and settlement growth in GLOF-exposed areas and underscores the importance of
considering exposure data in GLOF danger assessment. To address this in Bhutan, we
redefined dangerous glacial lakes by coupling flood characteristic modelling and downstream
exposure data. Accordingly, we have produced flood mapping and GLOF danger ranking for
278 glacial lakes along with comprehensive GLOF danger assessments for 274 local





government administrative units (LGUs). As a result, we classified lake130 (Thorthormi Tsho)
as a very high danger glacial lake in Bhutan, five lakes (lake93, lake251, lake262, lake276
and lake278) as high danger and 21 other lakes as moderate danger. Likewise, five
downstream LGUs were associated with very high GLOF danger while nine others were
associated with high GLOF danger.

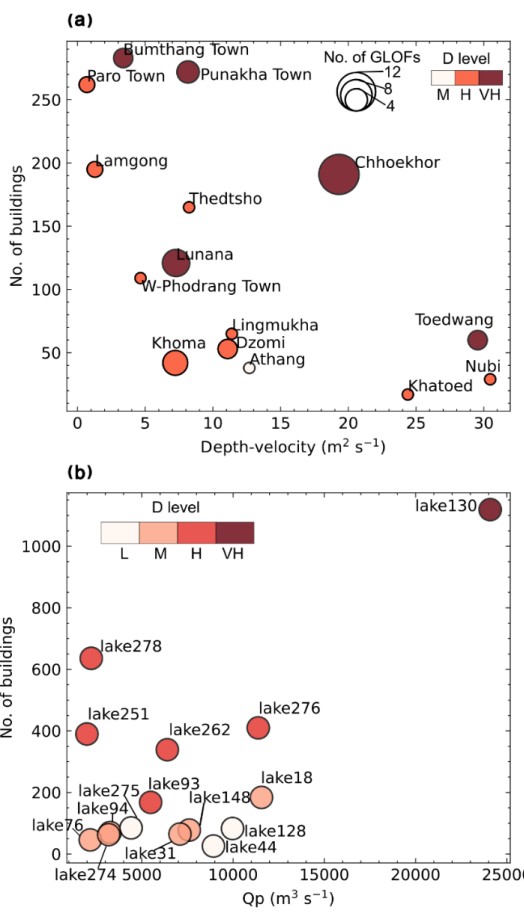

**Figure 7.** Dot plot illustrating the influence of GLOF magnitude and downstream exposure on
danger level computed in this study for (a) downstream LGUs and (b) individual lakes. In panel
(a), the GLOF magnitude is based on median depth-velocity across all the GLOFs that strike
at least one building in the LGU. In panel (b), we considered peak discharge ($Q_p$) as a proxy
for GLOF magnitude. In both the panels, colour associated with each dot indicates the GLOF
danger (d) level associated with each LGU or lake. The size of the dots in panel (a)
corresponds to number of GLOFs from various glacial lakes that impact the respective





community. For the better visualization, only top 15 dangerous glacial lake and top 15 LGUs
with GLOF danger are displayed here.
Our approach departs from many existing practices of identifying dangerous glacial lakes,
which are primarily based on the susceptibility of lakes to produce GLOF without regarding
the characteristics of settlements located downstream of the lakes (Rinzin et al., 2021; NCHM,
2019), in two ways: **1) Incorporation of GLOF Hydrodynamic characteristics**: we
considered flow velocity and flow depth, which are both primary components of the GLOF flow
that determine damage to the downstream elements (Federal Emergency Management
Agency, 2004). **2) Interaction of flow depth and velocity with downstream exposed**
**buildings:** we mapped potential downstream building damage associated with each GLOF
event based on the interaction between the depth-velocity and downstream at-risk elements.
By focusing on the interaction between flood magnitude and downstream exposed building,
our method classifies glacial lakes as dangerous only when their potential flood poses a threat
to downstream elements, making it a more practical and effective strategy for bespoke GLOF
risk reduction activities. For example, lake278 and lake251 are small and they produce
relatively small GLOFs with their estimate peak discharge approximately 2000 m$^3$ s$^{-1}$.
However, both were classified as high danger as the GLOF from these lakes impact hundreds
of downstream buildings (Fig. 7). Likewise, our approach assigns a higher GLOF danger
ranking to communities that are either affected by GLOFs from multiple lakes, impacted by
high-magnitude GLOFs, or have multiple buildings located within the GLOF inundation area,
whilst also considering the community's vulnerability. For example, Chhoekhor was identified
as having the highest GLOF danger in Bhutan because at least 191 buildings were potentially
impacted by GLOF from as many as 14 lakes in the basin. On other hand, other gewogs such
as Toedwang in Punakha also are classified as having very high GLOF danger, despite having
a comparatively low number (60) of potentially impacted buildings because these building
could be impacted by very high magnitude GLOFs in terms of depth and velocity (Fig. 7).
Our approach challenges traditional dangerous glacial lake assessments by redefining which
glacial lakes pose the greatest danger to the downstream settlements. As a result, we
identified three new high danger glacial lakes including, lake93 (Phudung Tsho), lake251, and
lake278 (Wonney Tsho), which are not recognized as dangerous glacial lakes by any of the
previous studies. Also, 53 of the previously identified 64 very highly susceptible to GLOFs
lakes (Rinzin et al., 2021) are categorized as low GLOF danger lakes. Conversely, 12 lakes
classified as low or very low GLOF susceptibility emerge as moderate to high danger in our
study (Rinzin et al., 2021). Likewise, nine of the dangerous lakes monitored by NCHM (six in
Punatsangchu basin, one each in Mangdechu, Chamkharchu and Kurichu basins) (National
Centre for Hydrology and Meteorology, 2019) are categorized as low danger in our study (Fig.
8). These discrepancies arise because we classified lakes as dangerous only if a potential
GLOF would affect a significant number of downstream buildings, whereas earlier studies
relied solely on geomorphological characteristics of the lakes and their surroundings (Rinzin
et al., 2021; NCHM, 2019) (Fig. 8). For example, lake278 in the Wangchu headwaters is
classified as high danger in our study because a potential GLOF could impact 636 buildings
across seven LGUs in Paro while the earlier studies considered this lake as safe as it does
not have geomorphological characteristics and lake condition to qualify as dangerous (Rinzin
et al., 2021).

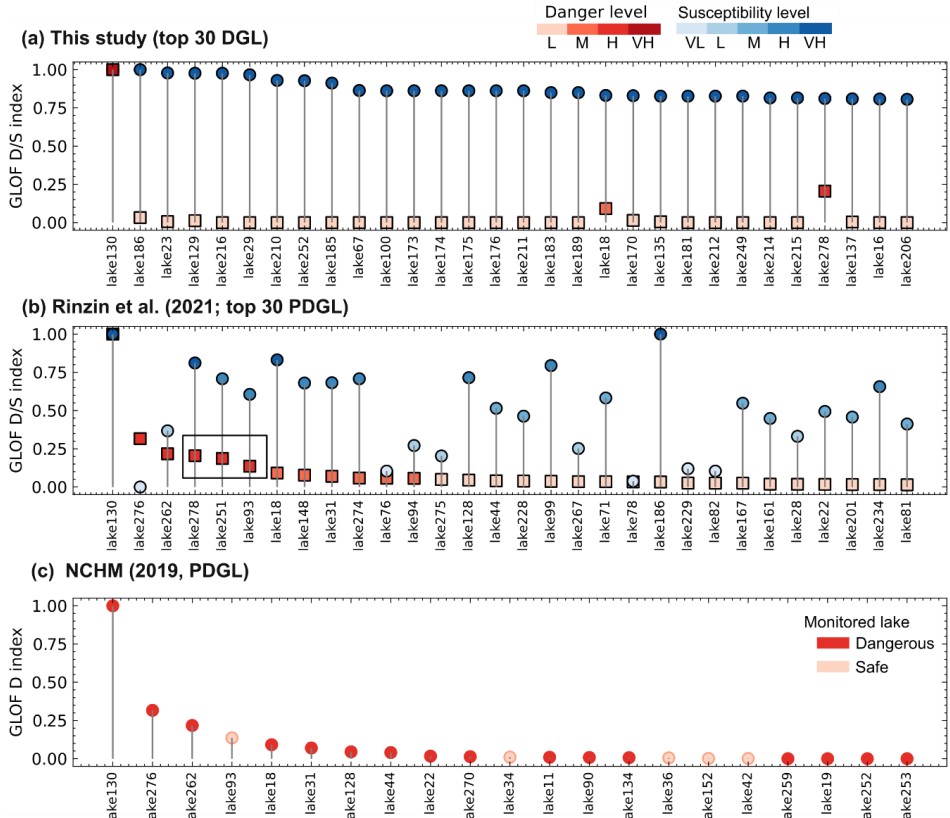


**Figure 8.** Comparison (a, b) GLOF damage index (DI) for top 30 dangerous glacial lakes
(DGL) calculated in current study and GLOF susceptibility score from Rinzin et al. (2021), and
(c) damage index (DI) for potentially dangerous glacial lakes (PDGLs) in Bhutan identified by
National Centre for Hydrology and Meteorology (2019). The black bounding box in panel (b)
shows the new dangerous glacial lakes identified in this study for the first time.




By ranking GLOF danger for all 274 LGUs, we discovered new GLOF risk hotspots such as
Paro town and Lamgong gewog (Paro, Khoma gewog [Lhuentse] and Chhoekhor gewog
[Bumthang]). GLOF danger in these places was previously not quantified and existing GLOF
early warning systems in Bhutan currently do not cover these high GLOF danger LGUs
(NCHM, 2021). We therefore recommend prioritizing monitoring of glacial lakes in Bhutan
based on high downstream exposure, rather than focusing on lakes selected solely based on
geomorphic susceptibility assessments. Specifically, Bhutan's glacial lake monitoring and
downstream risk mitigation efforts should expand beyond Lunana to include other high GLOF
danger lakes and vulnerable downstream settlement such as Paro Town, and Chhoekhor
gewog and gradually expanding to currently understudied areas in far eastern districts such
as Lhuentse, while emphasizing that higher granularity studies might be needed to guide
bespoke risk reduction efforts in these respective areas.
**5.2. Transboundary GLOF**
None of the transboundary lakes were classified as very high or high danger based on
potential GLOF impacts in Bhutan. This is because damage was minimal, mainly inundating
uninhabited parts of Bhutan, located in deep, inaccessible gorges. However, we identified
GLOF from four lakes in the Drangmechu basin (located in Arunachal Pradesh, India) and 11
in the Kurichu basin (located in the Tibetan Autonomous Region, China), which could
potentially impact several buildings in Bhutan.  Furthermore, GLOF from 20 lakes located in
the Indian and Chinese territories of the Himalaya enter Bhutan, although they do not impact
any building (Fig. 9). Identifying potential transboundary GLOFs is vital, given their potential
destructive power and long run out distances and challenges stemming from absence of
transboundary GLOF risk mitigation mechanism. For example, a recent GLOF from South
Lhoknak Lake in the Indian Himalaya has travelled over 300 km downstream, causing
significant damage in Bangladesh (Sattar et al., 2025). The absence of transboundary
cooperation for GLOF risk mitigation between Bhutan, China, and India complicates efforts to
monitor and manage such risks. Establishing regional cooperation is essential to enhance
early warning systems, facilitate data sharing, and implement coordinated risk reduction
strategies, thereby minimizing the potential damage from future transboundary GLOFs.
**5.3. Significance, limitations and the way forward**
Our approach of GLOF danger assessment using both flood magnitude and downstream
exposure data provides local authorities and relevant stakeholders with valuable information
to plan and prioritize wide ranging risk mitigation activities. These activities may target either
specific glacial lakes or downstream communities based on the danger index and level we
have provided, whilst also incorporating practical factors, such as resource availability and





logistical constraints. This study is particularly timely, as the Royal Government of Bhutan is
planning to modernize and expand its network of flood monitoring and GLOF early warning
systems (World Bank, 2024). This initiative, outlined in the roadmap for 2024–2034, aims to
develop multi-hazard warning services, aligning closely with the practical applications and
insights provided by our research. For example, our flood mapping and flow arrival time data
can be used to appropriately locate GLOF monitoring stations for early warning systems
(Wang et al., 2022). Likewise, some of the scattered buildings in LGUs such as Soe gewog in
Paro could be impacted by GLOFs within as little as 10 minutes. This short lead time means
it is practically not effective to install early warning systems for residents in these rapidly
affected areas. In such context, our flood extent mapping can effectively guide land-use zoning
and support informed decision-making for future development in these vulnerable locations.

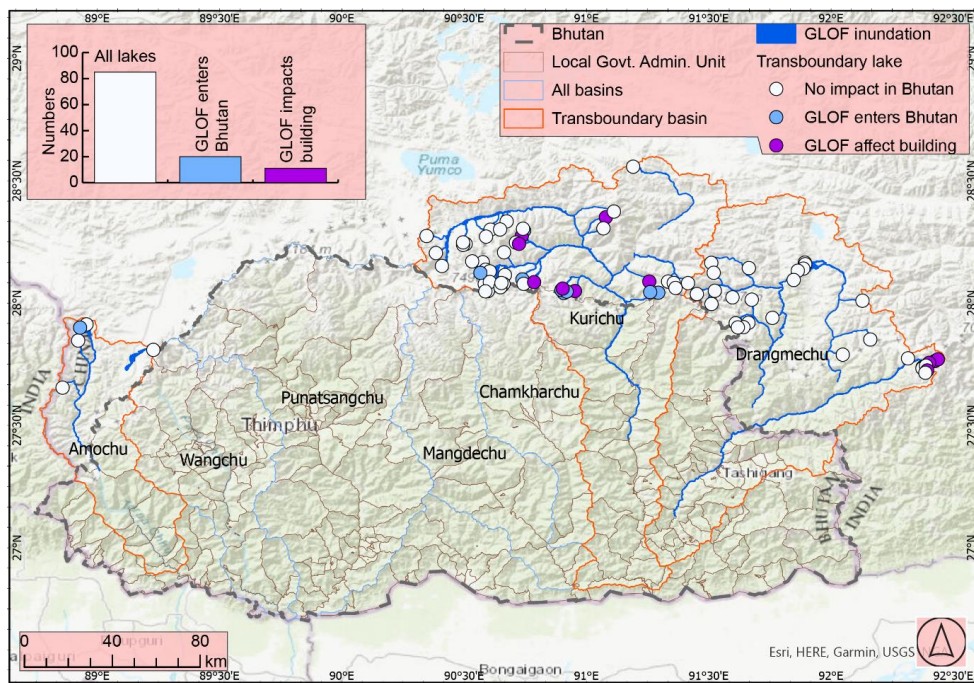


**Figure 9.** The map shows the impact of GLOF in Bhutan, which originates from lakes located
on the Chinese and Indian sides of Transboundary basins. The inset bar graph shows the total
lakes, lakes from which GLOF enters Bhutan and lakes from which GLOF impacts buildings
and other structures in Bhutan. The lake ID on the x-tick labels correspond to the ID on the
map. Base map image is the intellectual property of Esri and is used herein under license.
Copyright © 2025 Esri and its licensors. All rights reserved.



Our work establishes a baseline GLOF mapping and risk assessment in Bhutan. However, we
acknowledge that the magnitude of flood from glacial lakes will continue to evolve as glacier
retreat drives the expansion of existing lakes within a topographically constrained extent and
the formation of new lakes within the depressions left by the retreating glaciers (Zheng et al.,
2021b; Furian et al., 2022). Concomitantly, the downstream settlements within the GLOF-
prone areas are evolving, with population growth and infrastructure development leading to
increased GLOF exposure. The interplay of these factors means GLOF danger will likely
increase in the future and highlights the need for dynamic and regularly updated GLOF flood
mapping and risk assessments in the future.
We determined the minimum glacial lake area threshold (0.05 km$^2$) for GLOF modelling based
on the empirical evidence from the previous inventory (Shrestha et al., 2023; Komori et al.,
2012). However, it is important to acknowledge that glacial lake smaller than 0.05 km$^2$ have
also been known to produce GLOF event with a magnitude substantial to cause significant
downstream damage, particularly when it combines with other flood like meteorological flood
(Allen et al., 2015) or when the outburst flow entrains large amount of debris (Petrakov et al.,
2020; Cook et al., 2018). Thus, the future modelling effort should also consider smaller lakes
than the size threshold we considered here.
While we mapped all types of exposed elements located within the GLOF flow inundation
extent, our GLOF danger index is calculated solely based on the impact on number of exposed
buildings. This approach is grounded in the rationale that buildings represent the primary
places where people reside and are therefore the most direct proxy for population exposure.
However, critical infrastructure such as hydropower plants (e.g., in the Punatsangchu basin)
and the international airport in Paro (Wangchu basin), which are vital to the national economy,
were not included in our danger calculation. This omission stems from the considerable
challenges involved in accurately estimating the economic cost of potential damage to such
high-value infrastructure. When the GLOF intercepts hydropower dams, it can cause
overtopping, excessive sedimentation, outages, equipment damage leading to significant
revenue losses from the hydropower plants (Dunning et al., 2006) as well as cascading
impacts on the low-lying settlements (Sattar et al., 2025). Likewise, damage to the crucial
infrastructure such as Paro international airport will hinder relief effort after the GLOF disaster
delaying the recovery and escalating overall loss and damage. Therefore, future study should
also consider absolute economic impact of GLOF to aid relevant stakeholders and
policymakers in developing appropriate strategies to mitigate risks to vital infrastructure.
Looking forward, the glacial lake dataset can be updated using wide-ranging open-access
remote sensing imagery. Similarly, platforms such as OpenStreetMap, which leverage crow-





sourced data and are frequently updated, present a valuable resource for mapping evolving
downstream buildings and other structure data. Likewise, hydrodynamic modelling for multiple
glacial lakes with freely available and user-friendly models such as HEC-RAS is increasingly
becoming feasible with the recent development in artificial intelligence and cloud-based
computing platforms like Flood Platform (https://www.floodplatform.com/) which enable
integrating products from varied flood simulations/models into a common framework. We will
develop web portal, which hosts glacial lake data and flood maps, serves as a valuable
resource for periodic updates to flood damage assessments. By integrating up-to-date glacial
lake flood magnitude information with evolving downstream exposure data, this platform can
provide valuable information for informed decision-making and proactive risk management,
such as tailored early warning systems and land use management and development.

## 6. Conclusion

Glacial lakes, which are growing in number and areas in the mountains globally, pose a serious
GLOF threat to the communities living downstream of them. However, the destruction and
damage caused during the GLOF events are not only a function of lake drainage magnitude
but also depend on their interaction with downstream exposed elements. Despite this,
traditional approaches to assessing danger posed by glacial lakes have been mainly based
on the likelihood and magnitude of a lake to produce GLOFGLOF and often disregard the
potential downstream impact. To address this gap, this study redefines the classification of
dangerous glacial lakes in Bhutan (one of the high GLOF risk countries globally) by combining
GLOF hydrodynamic characteristics mapping and downstream exposed buildings.
This study produced GLOF hydrodynamic characteristics for all glacial lakes in Bhutan which
are greater than 0.05 km$^2$ and located within the 1 km of glacier terminus. The analysis
revealed that approximately 22399 people, 2613 buildings, 270 km of road, 402 bridges and
20 km$^2$ of farmland are exposed to GLOF in Bhutan. A GLOF damage index was developed
by combining flood mapping data with downstream exposure metrics, enabling the ranking of
glacial lakes based on their potential danger. Thorthormi Tsho was identified as the most
dangerous glacial lakes in Bhutan. Furthermore, we identified five additional glacial lakes as
having high GLOF danger, two of which are in head water of Wangchu, neither included in
previous study and nor monitored by existing early warning system in Bhutan. Among these
dangerous glacial lakes, three of them are newly identified dangerous glacial lakes (lake251,
278 and lake93) in the current study.
For the first time, this study provides GLOF danger ranking for 20 districts and 274 local
government administrative blocks (gewogs and towns) [LGUs] in Bhutan. In addition to the



previously identified high GLOF danger gewogs and towns, we have identified six additional
LGUs with similarly high GLOF dangers. These include Chhoekhor and Bumthang town in
Bumthang, Paro town and Lamgong in Paro, Nubi in Trongsa and Khoma in Lhuentse districts.
Most strikingly, some downstream LGUs such as Paro town and Lamgong gewog in Paro are
not covered by the existing Bhutan early warning system, highlighting significant gaps in
existing risk mitigation efforts.
This study underscores the criticality of incorporating flood mapping and downstream
exposure and vulnerability data when defining GLOF dangerous lake and assessing
downstream risk. For Bhutan, the findings emphasize the urgent need to expand and
strengthen GLOF risk mitigation strategies, including the enhancement of early warning
systems and the implementation of targeted interventions in newly identified high-risk areas.
These measures are essential to safeguarding vulnerable communities and infrastructure from
the escalating threat of GLOFs in the context of ongoing climate change and glacial retreat.
**Acknowledgement**
This work was supported by the Natural Environment Research Council (NERC)- funded
IAPETUS Doctoral Training Partnership [IAP2-21-267].
**Code and data availability**
The HEC-RAS 2D model we used here for simulating glacial lake outburst modelling can be
accessed at: https://www.hec.usace.army.mil/. The AW3D30 DEMS used here can be
downloaded from the OpenTopogragphy at: OpenTopography - Find Topography Data. Bhutan
2017 housing and census data can be downloaded from National Statistical Bureau of Bhutan
at https://www.nsb.gov.bt/. Landover and landuse data used in this study can be accessed at:
https://rds.icimod.org/.    The    OpenStreetMap    data    can    be    assessed    at:
https://www.openstreetmap.org/relation/184629. GLOF hydaulic data for each glacial lake will
be made available through web portal with publishing of this ariticle.
**Supplement**
The supplement related to this article is available online at:
**Author contributions**
SR, SD and RC conceptualized the study. SR undertook data analysis, visualization and wrote
original draft. SD and RC secured the funding, supervised and contributed equally to the work.



SA, AS and SW reviewed and edited the manuscript. All authors contributed to the final
mauscript.
**Competing interests**
The contact author has declared that none of the authors has any competing interests.



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
