# Peer review of "Redefining dangerous glacial lakes in Bhutan by integrating hydrodynamic flood mapping and downstream exposure data"

_EGUsphere, 2025_

## Author Comment (AC1)

**Reviewer #2**

This study models and assesses the impacts of potential future GLOFs in Bhutan and takes into account elements at risk. The methodology is innovative and goes beyond susceptibility / hazard assessment. It is important to acknowledge the amount of work behind it and appreciate implications for the development of GLOF EWS in Bhutan. I have read the study with great interest and have number of questions / comments to it:

Many thanks, Dr. Adam Emmer, for your positive review of your manuscript. Your feedback has greatly helped improve our manuscript. We are pleased to submit the following point-by-point response to your comments. Our responses are highlighted in blue colour for more visibility.

L1: how do the authors know these are the "most likely" scenarios without looking at triggers and dam properties of individual lakes?

We inferred the most likely scenario based on the median value from drainage volume data of historical GLOF events in HMA. However, considering that the median value does not necessarily represent the most likely scenario, we have now corrected the statement as follows:

"Here we redefined dangerous glacial lakes by considering the impact of a simulated GLOF scenario from each lake on downstream exposed elements at risk."

L37: water addition is not the driver; expansion of existing glacial lake basins / formation of new basins is

Amended as "accumulation of meltwater on a newly exposed depression left by a retreating glacier".

L47: is there any study actually showing that moraine-dammed lakes are closer to settlements? I argue that the reason explaining this observation is rather linked to specific GLOF mechanisms from moraine-dammed lakes (<a href="https://www.nature.com/articles/s44221-024-00254-1">https://www.nature.com/articles/s44221-024-00254-1</a>)

Acknowledged and amended as follows:

"This is because moraine-dammed lakes events are high magnitude yet episodic, making them highly unpredictable for the downstream settlements. Likewise, moraine-dammed lakes are located in densely populated areas such as the HMA, Andes and Alps, in contrast to other types of glacial lakes, such as ice-dammed or supraglacial ponds/lakes (Emmer, 2024)."

L73: I suggest to use established terminology - susceptibility, hazard, vulnerability and risk

Thanks for the suggestions. The suggested terminologies are used in all appearances.

L91: "hydrodynamics" do not interact, GLOFs do interact

**Amended as suggested.**

L108: the problem usually is not the susceptibility assessment itself, but lake size threshold used.

Thanks for pointing this out. Considering that our work does not adequately address the GLOF risk from the smaller lakes, as pointed out by both reviewers, we are avoiding this statement here. However, we have discussed this limitation adequately in the revised manuscript (Lines 699 to 706).

L114: using the 50,000 m2 threshold is arbitrary; there are examples of damaging or even deadly GLOFs from much smaller lakes (e.g. the Independencia/Huaráz GLOF earlier this year coming from approx. 10,000 m2 lake)

Thanks for pointing this out. Our selection of this size threshold was first motivated by the computational difficulties involved in modelling the hydrodynamics of all lakes, considering a threshold of 10,000 m². This is because employing such a threshold would mean modelling over 1,000 more glacial lakes. But, our selection was also based on the previous GLOF data available at the HMA scale, where the known area of pre-event lakes was greater than 0.05 km² (Shrestha et al., 2023). Having said this, we acknowledge the reviewer's concern and do not discount the risk posed by smaller glacial lakes. However, given the emphasis of our study on downstream impact, we believe that our study covers downstream settlements and nearly all buildings located below all the glacial lakes in Bhutan (including smaller ones), although the risks associated with smaller glacial lakes (<0.05 km²) are not covered. We believe that this consideration is important, as a glacial lake's likelihood is not predictable based on its size (be it small or large) and its surrounding conditions. Nevertheless, we have explicitly discussed the importance of considering small lakes in the discussion section 3.2 as follows (Lines 708 to 715):

"We determined the minimum glacial lake area threshold (0.05 km²) for GLOF modelling based on the empirical evidence from the previous inventory (Shrestha et al., 2023; Komori et al., 2012). However, it is important to acknowledge that glacial lakes smaller than 0.05 km² have also been known to produce GLOF events with a magnitude substantial enough to cause significant downstream damage (Sattar et al., 2025a), particularly when they combine with other forms of floods like meteorological floods (Allen et al., 2015) or when the outburst flow entrains a large amount of debris (Petrakov et al., 2020; Cook et al., 2018). Thus, the future modelling effort should also consider smaller lakes than the size threshold we considered here."

L124: please provide a link to this portal

This web portal has been in the process of development. We will provide the link in the process of publication.

L139: Rinzin et al. 2021 also used 50,000 m2 threshold in their assessment, meaning that the danger of GLOFs from small lakes (regardless their proximity to settlements or infrastructure) is systematically overlooked.

Thanks for pointing this out. We now amended as follows in the revised manuscript:

"Of these 64 glacial lakes greater than 0.05 km2 were identified as highly or very highly susceptible to producing GLOF in the future based on geomorphological conditions such as topographical potential for avalanching into the lake (Rinzin et al., 2021)"

Fig. 1: elevation is not readable since the country is coloured in green; consider using the same format for all coordinates (90°E vs. 90°30'E)

**Amended as suggested**

L177: yes, because GLOFs from smaller lakes and / or farther in the past tend to happen unnoticed

Thanks. We have now removed this statement in the revised manuscript to acknowledge the risk from the smaller lakes.

L183-185: this is a common hazard-oriented filtering; considering the points made in the intro, I wonder why a distance from settlements / infrastructure is not considered as a primary filter rather than lake size and distance from glaciers?

We did not consider the distance from the settlement since our GLOF risk assessment is based on the downstream impact using hydrodynamic mapping. We believe that large dangerous glacial lakes with great GLOF reach distance would be excluded from the selection if the distance to the settlement criteria is considered. Also, our assessment being based on hydrodynamic modelling, we believe that distance to settlement is taken care by the flood reach distance we generated for each lake.

Fig. 2: it is not clear to me from the figure how a damage level which is calculated for each cell of the mesh by multiplying max. velocity with max. depth is translated into one damage index for each lake? A normalized sum? It is important to acknowledge that normalisation is sensitive to outliers (very large lake130 is in order of magnitude larger than any other lake in the dataset); DV is not defined correctly - L1 is a subset of L2 (<5 is a subset of <10), 10 does not fall to any category; please check and revise

We like to clarify that our classification system is based on the natural jenk system, which takes care of large disparities in damage values associated with glacial lakes. For this reason, we have classified only one lake, that is Thorthormi Tsho, as a very high danger glacial lake. However, acknowledging the large difference between a very high danger and high danger glacial lake, we have added the following discussion (in lines 604 to 611):

"We classified only one lake (Thorthormi Tsho) as a very high danger glacial lake. This is because, at about ~4.3 km² in size, Thorthormi Tsho is the largest glacial lake, approximately double the size of the next largest glacial lake in Bhutan. Moreover, Thorthormi Tsho is located in the Punatsangchu basin, which is among Bhutan's most populated basins, resulting in exposure of up to 1,119 buildings, again, an order of magnitude more exposure than the next lake associated with the highest exposure. These findings further highlight the importance of

strengthening risk mitigation measures for Thorthormi Tsho and the affected downstream settlements (NCHM, 2019; Rinzin et al., 2021)"

Also, we would like to clarify that we classified damage level as follows: L1 (DV =  $5 \text{ m}^2\text{s}^{-1}$ ), L2 (DV =  $5-10 \text{ m}^2\text{s}^{-1}$ ) and L3 (>10 m2s-1).

L212-213: this is a justified approach but it does not give the most likely volume scenario (which would have to be treated lake by lake, considering dam geometry and properties and potential triggers)

We acknowledge the reviewer's comments about the variability in drainage volume during the GLOF event due to various factors. However, GOF risk assessment involving large numbers of glacial lakes across the entire Bhutan considered for this study means conducting that level of detailed assessment is difficult and not feasible. Having said this, acknowledging the reviewer's comments, we added the following in the corresponding lines (209 to 213):

"The amount of water drained during a GLOF event depends on numerous factors, such as dam geometry and composition, lakebed topography, and potential triggers. While consideration of all these factors is crucial for a detailed impact assessment of a particular glacial lake, constraining the volume based on these detailed attributes is highly challenging for a study involving numerous glacial lakes"

L221: most GLOFs do not originate from moraine-dammed lakes (as mentioned earlier in the Intro)

Thanks for the correction. It is now amended as "Most GLOFs from moraine-dammed lakes start from dam breaching"

L246: justification of parameters with the use in previous studies should be accompanied by performance evaluation (from the original study or elsewhere), otherwise it is prone to proliferate the use of possibly irrelevant parameter values.

Thanks for pointing this out. While we acknowledge the requirement of such a performance evaluation, we would like to clarify that we did not consider a performance evaluation for this, considering that our study is to provide an overview of GLOF hazard and risk in Bhutan instead of providing an accurate risk score or hazard mapping. However, considering the reviewer's suggestions, we have added the following in line 252:

Chow (1959) suggested Manning's n between 0.040 to 0.070 for the river channel bed with large boulders and cobbles, which closely characterise the river channels in the Bhutan Himalaya. Thus assigned Manning's n value of 0.06 which has been used in GLOF modelling in Bhutan previously (Maurer et al., 2020).

Furthermore, we will now add sensitivity analysis considering the variability of Manning's n (between 0.040 and 0.070) and lake size variation between 0.01 and 4.5 km². For lake size, we considered four scenarios: >0.05 km², 0.05 to >0.1 km², 0.1 to 1 km², and >1 km² in line with the size category we considered for calculating drainage volume. This sensitivity analysis will be conducted within the same basin to uniformly assess downstream impacts. Results of

the sensitivity analysis will be added, and their implications will be discussed in the revised manuscript.

L246: again, better justification for the use of this value would be with the values recommended for different Channel types - (https://www.fsl.orst.edu/geowater/FX3/help/8\_Hydraulic\_Reference/Mannings\_n\_Tables.ht m) I think 2b fits well most of the GLOF streams

Thanks for the suggestions. We amended as mentioned above.

L269: same as above - previous use doesn't guarantee satisfactory performance and suitability

Thanks for pointing this out. We revised the sentence as follows in the revised manuscript:

"Considering that this study is mainly aimed at providing GLOF hazard and risk overview at the Bhutan scale, all other computational parameters were maintained at the default setting."

L283-287: population density is suitable for hazards affecting large continuous areas (e.g., earthquakes, heat waves) but GLOF impact areas are very localized and the approach ignores spatial patterns of population distribution; since the authors have detailed data about individual buildings, I was wondering why not to estimate exposed population as a function of built-up area, instead of using population density multiplied by inundation area?

Thanks for pointing this out. As suggested, we have calculated the exposed population as a function of built-up area in the revised manuscript.

L314: what is the meaning of these values 8 and 10?

The values 8 and 10 are the arbitrary numbers assigned to buildings in villages and towns following Carrivick and Tweed (2016). This is based on the assumption that buildings in the towns have higher values than those in the rural areas.

However, assuming that buildings in the towns are also stronger compared to those in the rural areas. Also, considering that this study provides only a first-order risk assessment, we will assign damage value to all the buildings. Any change in the results and conclusion will be reported in the revised manuscript.

L330: Is there any specific reason why the values were categorized? And any specific reason why into these three categories? Categories blur differences. Why not normalize the values similarly to the SE indicators? It would be good to show at least the histogram of values so it is easier to understand how the values are distributed and how the dataset is split into categories.

Thanks for the suggestion: We have made the amendments as suggested in the revised manuscript, and a histogram plot showing the distribution of DV has been added to the supplementary material.

L331: please check if the first category really includes 0? categories are not defined correctly - 5 falls to both L1 and L2

Thanks for pointing this out. The categories are meant to be as follows: L1 (0 to <5), L2 (>5 to 10) and Level 3 ( $> 10 \text{ m}^2 \text{ s}^{-1}$ ). We will amend this in the revised manuscript.

L336: equation (iii) and (iv) can be merged? the sum function needs definition of i (typically from 1 to n, where n is the number of impacted grid cells)

These two equations have been merged as suggested in the revised manuscript

L344: what is a gewog? please give approximate translation or explanation

Gewog is the sub-district block. It has been defined in the revised manuscript.

L373-374: the selection of SE indicators is not justified; how sensitive is the outcome to changes in inputs? If 18 indicators are available, it does not necessarily mean that the set of 18 indicators is the best to use. How sensitive is it to outliers?

These SE indicators were used considering that these indicators proxy of people's response capacity and if otherwise vulnerability. The selection of these parameters is based on our local knowledge and previous literature. Because we use multiple SE indicators and they were all normalized to a uniform value, we believe that they are least impacted by the outliers, as shown by the supplementary figure S4.

L375: all indicators should be relative values before they are standardized, otherwise large LGUs are getting higher values and so higher normalized values; for example, it is stated that the measure of the indicator "Reliable source of energy for lighting" is defined as "Households with a main source of energy for lighting as electricity"; if so, this is the absolute value which is meaningless to normalize unless it is related to the number of households in individual LGUs, i.e. "Proportion of households with a main source of energy for lighting as electricity"

We would like to clarify that all the indicators are proportionate to either the total population or households in the LGUs. To make this clear, we have amended Table 1 in the revised manuscript

L380: building damage is not only a function of hydrodynamics factors but construction type and construction quality factors too

Amended the line as: Building damage from GLOF is a function of hydrodynamic factors such as depth and velocity, and structural integrity of the buildings.

L386-389: please consider elaborating this earlier - Intro or study area section; which lakes they found hazardous?

Thanks for the suggestions: We have added the following in the study area section (lines 143 to 146):

"Likewise, using similar criteria, NCHM has identified 17 potentially dangerous glacial lakes NCHM, 2019) based on the earlier assessment by ICIMOD (Mool et al., 2001). Of these dangerous glacial lakes, the majority (n=9) are located within the Phochu basins, the headwater of Punatsangchu."

L392: how?

Thanks. To make it clearer, we have now modified as follows: To achieve this, we first overlayed all monitoring stations in Bhutan using ArcGIS

L396: missing verb in the sentence?

Thanks. The verb 'was' added.

Table 2: this shows the downside of using population density and ignoring spatial patterns of settlement location (see my comment above); comparing Bumthang Town and Lunana, the mean population of a mean building ranges from 0.71 person per building (Lunana) to 20.3 persons per building (Bumthang), i.e. in two orders of magnitude.

Thanks, now the population is calculated as a function of built-up density as suggested by the reviewer and Table 2 is modified accordingly in the revised manuscript.

L402: the structure in methods starts with hazard, followed by exposure and vulnerability; it is contraintuitive that Results section starts with Impact and exposure; I suggest to unify the structure in methods and results

Many thanks for this suggestion. We have now unified the flow between the method and results. In both cases, it starts from hazard followed by exposure, vulnerability and risk. Accordingly, we added the following section under the result section: "GLOF characteristics"

L433: and SE vulnerability indicators

SE vulnerability indicators have been added

L435-440: consider moving to methods

We have moved the method components in this sentence to the method section (line 345 to 348). Now the sentence reads as:

"Among other lakes, with the highest Di, Thorthormi Tsho in the Punatsangchu basin emerged as the most dangerous high danger lake in Bhutan (Fig. 3). Five other lakes are identified as high danger, which are distributed across the Wangchu (2), Chamkharchu (2), and Punatsangchu (1) basins."

L445-446: can any of the lakes located in Bhutan generate transboundary flood to any of the neighbouring countries?

This has now been discussed in section 5.2 (line 661 to 666) as follows.

"The modelled GLOF from all lakes in Bhutan attenuate before it crosses the international border between Bhutan and India. However, we acknowledge that some of the GLOF events in the future can impact settlements along transboundary river flood plains in India, especially under the worst case scenario and when their flow get amplified by the addition of material along the flow path such as from landslide deposit (Cook et al., 2018) and hydropower dam (Sattar et al., 2025b)."

Fig. 3: why is building count expressed per km2?

Building count is expressed in km2 as we summarized the number of buildings in every one km2 along the flood plains for the visualization purpose.

Fig. 4: building damage level in shades of blue is not very intuitive

Thanks. We have now amended this in the revised manuscript

Fig. 5: flow arrival time is shown on x axis, right? What is the difference to flow arrival time indicated by the color of a circle? For example, why is the arrival time to Bumthang (45) 3 hours when reading from the x axis but 10 hours according to the color scale?

Thanks for the careful spot. We have now corrected the legend for the flow arrival time.

L523: how many residents in GLOF exposed buildings?

Amended as suggested.

L549: escalating exposure and so risk

Amended as suggested

L594: it does not challenge but expands on more common GLOF hazard assessment studies

Amended as suggested

L597-611: yes, but this is not contradicting or surprising that there are differences, is it? These are different approaches used for different purposes; one says which lakes are more or less likely to produce a GLOF while the other says which lakes are more or less likely to produce a damage in case of a GLOF

Thanks for the suggestion. As we used the term 'expand' as suggested above in the revised manuscript, we feel that this concern is now addressed.

L622-624: it is not one or the other; all components (h, e, v) need to be taken into consideration in GLOF risk management and I suggest to reword this sentence

Amended as follows: "We therefore recommend prioritizing monitoring of glacial lakes in Bhutan based on all components of risk (hazard, exposure and vulnerability), rather than

focusing on lakes selected solely based on geomorphic susceptibility assessments, which takes care of only physical hazard."

L631: lakes located outside Bhutan's border; how about transboundary GLOF danger originating in Bhutan?

Thanks for pointing this out. This has now been discussed in section 5.3, as mentioned earlier

L647: please consider structuring this sub-section into sub-sub-sections

Thanks. Amended as recommended in the revised manuscript.

L647: for the future work, please also highlight the need for developing site-specific GLOF scenarios that consider different triggers, dam types and geometries (which is (understandable) source of uncertainty in this country-wide study.

Thanks, we have added the following lines in the discussion (lines 719 to 730):

Our drainage volume and peak flow calculations are based on empirical equations and the previous GLOF events, with scarcely documented detailed characteristics (Shrestha et al., 2023). Employing such proxy parameters is reasonable for this study as our aim is to provide overview of GLOF risk in Bhutan based on the downstream impact. The modelled GLOF scenarios for each lake are directly comparable, enabling an assessment of the overall and relative levels of danger, and representing a moderate scenario. We recognize however, that this is just one set of scenarios. Due to time and computational constraints, it was not feasible to simulate all potential variations. Future studies focusing on the detailed impact of specific glacial lakes or on specific downstream communities must be grounded on site-specific scenarios informed by situational triggering factors and dam composition and geometry. The study should also consider the site-specific worst-case scenario, considering the future climatic conditions.

L647: please comment on the sensitivity of the assessment procedure to outliers (see my comment above); please comment on the selection of the SE indicators and categorisation vs. normalisation that are used concurrently in different steps of the procedure.

Thanks for the suggestion. Firstly, regarding the uncertainty stemming from outliers associated with glacial lake size, we have already addressed this in earlier comments.

Regarding SE indicator, we believe that there is no significant uncertainty from normalization of SE indicators as they are all considered based on the proportion of households and population. But we agree on the uncertainty associated with the selection of SE indicators. So, we have added the following lines in the discussion under section 5.3 (lines 747 to 756):

"The socio-economic indicators used here are the best available census data at the finest granularity in Bhutan. These indicators represent people's capability to respond to and recover from not only GLOF but also any natural or man-made hazards (Cutter et al., 2003). However, these indicators do not necessarily represent people's specific vulnerability to GLOF as it also depends on other factors such as prior experience of natural hazard (Foundation, 2024). For

example, we classified Lunana gewog as the most vulnerable gewog based on these socioeconomic indicators; however, how their prior experience influences their response capability remains beyond the scope of this study. Future study focusing on specific downstream settlement or impact of a particular glacial lake should also consider the broader implications of vulnerability and resilience."

L723: GLOFGLOF

**Amended**

L729: "approximately" should not be followed by very precise (though highly uncertain) numbers

Thanks. The sentence is now amended as: The analysis revealed that over 20,000 people, 2600 buildings, as well as other infrastructure (such as roads and bridges) and farmland are exposed to GLOF in Bhutan

Fig. 9: consider unifying graphical style with other maps in the study (please remove pink background from the legend)

Thanks. Uniform graphical style is used across all figures in the revised manuscript.

Table S1: the distribution of normalized danger index values is anomalous; removing one outlier of the lake 130 would likely generate way different results in terms of what is shown in Fig. 8a - please discuss this (see my comment above)

Thanks. This has now been discussed.

Overall, this is a state-of-the-art country-wide GLOF risk assessment study with some innovative methodological aspects. The manuscript would benefit from the use of standard risk terminology. Parts of the methodology require better justification / discussion. I recommend moderate revisions.

Thanks for your positive review of our manuscript. We have standardised the risk terminology and addressed the concern raised regarding our methodology throughout the manuscript.

The following references are used to address the comments from the reviewers.

(NCHM), N. C. f. H. a. M.: Resport on the rapid assessment of Thorthormi lake and restoration of automatic water level sensor for the GLOF early warning system, National Centre for Hydrology and Meteorology, 2019.

Allen, S. K., Rastner, P., Arora, M., Huggel, C., and Stoffel, M.: Lake outburst and debris flow disaster at Kedarnath, June 2013: hydrometeorological triggering and topographic predisposition, Landslides, 13, 1479-1491, 10.1007/s10346-015-0584-3, 2015.

Carrivick, J. L. and Tweed, F. S.: A global assessment of the societal impacts of glacier outburst floods, Global and Planetary Change, 144, 1-16, 10.1016/j.gloplacha.2016.07.001, 2016. Cook, K. L., Andermann, C., Gimbert, F., Adhikari, B. R., and Hovius, N.: Glacial lake outburst floods as drivers of fluvial erosion in the Himalaya, Science, 362, 53-57, doi:10.1126/science.aat4981, 2018.

- Cutter, S. L., Boruff, B. J., and Shirley, W. L.: Social Vulnerability to Environmental Hazards\*, 84, 242-261, https://doi.org/10.1111/1540-6237.8402002, 2003.
- Emmer, A.: Understanding the risk of glacial lake outburst floods in the twenty-first century, Nature Water, 2, 608-610, https://doi.org/10.1038/s44221-024-00254-1, 2024.
- Foundation, L. s. R.: World Risk Poll 2024 Report: Resilience in a Changing World., https://doi.org/10.60743/C0RM-H862, 2024.
- Komori, J., Koike, T., Yamanokuchi, T., and Tshering, P.: Glacial Lake Outburst Events in the Bhutan Himalayas, Global Environmental Research ©2012 AIRIES, 16, 12, 2012.
- Maurer, J. M., Schaefer, J. M., Russell, J. B., Rupper, S., Wangdi, N., Putnam, A. E., and Young, N.: Seismic observations, numerical modeling, and geomorphic analysis of a glacier lake outburst flood in the Himalayas, Science Advances, 6, eaba3645, doi:10.1126/sciadv.aba3645, 2020.
- Mool, P. K., Wangda, D., Bajracharya, S. R., Joshi, S. P., Kunzang, K., and Gurung, D. R.: <Inventory of Glaciers, Glacial Lakes and Glacial Lake Outburst Floods-Bhutan.pdf>, International
- Centre for Integrated Mountain Development (ICIMOD), Kathmandu, Nepal, 2001.
- National Centre for Hydrology and Meteorology, N.: Reassessment of Potentially Dangerous Glacial Lakes in Bhutan, National Centre for Hydrology and Meteorology, Royal Government of Bhutan, National Center for Hydrology and Meteorology, Royal Government of Bhutan, PO Box: 2017, Thimphu, Bhutan, 54, 2019.
- Petrakov, D. A., Chernomorets, S. S., Viskhadzhieva, K. S., Dokukin, M. D., Savernyuk, E. A., Petrov, M. A., Erokhin, S. A., Tutubalina, O. V., Glazyrin, G. E., Shpuntova, A. M., and Stoffel, M.: Putting the poorly documented 1998 GLOF disaster in Shakhimardan River valley (Alay Range, Kyrgyzstan/Uzbekistan) into perspective, Science of The Total Environment, 724, 10.1016/j.scitotenv.2020.138287, 2020.
- Rinzin, S., Zhang, G., and Wangchuk, S.: Glacial Lake Area Change and Potential Outburst Flood Hazard Assessment in the Bhutan Himalaya, Frontiers in Earth Science, 9, 10.3389/feart.2021.775195, 2021.
- Rinzin, S., Dunning, S., Carr, R. J., Sattar, A., and Mergili, M.: Exploring implications of input parameter uncertainties in glacial lake outburst flood (GLOF) modelling results using the modelling code r.avaflow, Natural Hazards and Earth System Sciences, 25, 1841-1864, 10.5194/nhess-25-1841-2025, 2025.
- Rinzin, S., Zhang, G., Sattar, A., Wangchuk, S., Allen, S. K., Dunning, S., and Peng, M.: GLOF hazard, exposure, vulnerability, and risk assessment of potentially dangerous glacial lakes in the Bhutan Himalaya, Journal of Hydrology, 619, 10.1016/j.jhydrol.2023.129311, 2023.
- Sattar, A., Haritashya, U. K., Kargel, J. S., and Karki, A.: Transition of a small Himalayan glacier lake outburst flood to a giant transborder flood and debris flow, Sci Rep, 12, 12421, 10.1038/s41598-022-16337-6, 2022.
- Sattar, A., Emmer, A., Lhazom, T., Rai, S. K., and Azam, M. F.: Flood risk from small mountain lakes, Communications Earth & Environment, 6, 10.1038/s43247-025-02758-4, 2025a.
- Sattar, A., Goswami, A., Kulkarni, A. V., Emmer, A., Haritashya, U. K., Allen, S., Frey, H., and Huggel, C.: Future Glacial Lake Outburst Flood (GLOF) hazard of the South Lhonak Lake, Sikkim Himalaya, Geomorphology, 388, 10.1016/j.geomorph.2021.107783, 2021.
- Sattar, A., Cook, K. L., Rai, S. K., Berthier, E., Allen, S., Rinzin, S., Van Wyk de Vries, M., Haeberli, W., Kushwaha, P., Shugar, D. H., and et al.: The Sikkim flood of October 2023: Drivers, causes and impacts of a multihazard cascade, Science, 0, eads2659, 10.1126/science.ads2659, 2025b.
- Shrestha, F., Steiner, J. F., Shrestha, R., Dhungel, Y., Joshi, S. P., Inglis, S., Ashraf, A., Wali, S., Walizada, K. M., and Zhang, T.: HMAGLOFDB v1.0 a comprehensive and version controlled database of glacier lake outburst floods in high mountain Asia, Earth Syst. Sci. Data Discuss., 2023, 1-28, 10.5194/essd-2022-395, 2023.

---

## Author Comment (AC2)

**Reviewer#1**

The study by Rinzin et al. analyses the downstream exposure and vulnerability of infrastructure, buildings and people to glacial lake outburst floods in Bhutan. Their analysis relies on a dataset >200 glacial lakes, a globally available digital elevation model and OSM data. Using HEC-RAS, the authors simulate a scenario for each lake and compare flooding extents, depths and velocity to the locations of the elements at risk.

In general, this is a well-conceived study that leverages hydrodynamic modelling to address some of the weaknesses of previous studies that made quite simplifying assumptions about flood wave propagation and the extent of their impact. However, there are still a few issues with the study which I will outline below. All in all, I recommend major revisions before the manuscript should be published in NHESS.

Many thanks for providing valuable feedback and a positive review of our manuscript. The comments were highly valuable, which helped us improve the quality of our manuscript. We are pleased to provide our response in the following, where our responses are highlighted in blue fonts for better visibility

**Major comments:**

The parameter choice relies on previously published data (flood volume). However, the choice of parameters does not consider the variability of this data, but rather takes point estimates. For example, the choice of using the median of reported percentages of drainage volume is considered the "most likely flood volume" (L 212). However, if you have a bimodal distribution of partial drainage volumes, then the median is not the most likely flood volume. Thus, it may be useful to not pick out the median scenario, but one that is at the upper end of the distribution, thus giving more weight to extreme scenarios. Same is true for the volume-area relation that may only represents an average of the breadth of possible scenarios. Schwanghart et al. (2016) showed that results of GLOF modelling are not sensitive to uncertainties in the V-A relation for large lakes, but that these uncertainties matter for smaller lakes. I acknowledge that the study already comprises many simulations with quite a heavy computational load. However, it should be at least discussed that the current approach lacks a consideration of the large variability of possible outburst scenarios and that average scenarios may not capture the worst-case scenarios.

We thank the reviewer for raising this important concern. We recognise that this is just one set of scenarios, but we cannot run them all due to time and computing constraints. We therefore pick the median scenario, so they are directly comparable, to provide an estimate of the overall, relative danger and to give a moderate scenario, so we are not essentially modelling outliers. We are now not using the term 'most likely scenario' but a more specific, 'medium scenario' in all appearances in the revised manuscript. We have also addressed this limitation and its implications in the discussion section in the revised manuscript (lines 716 to 724).

"Our drainage volume and peak flow calculations are based on empirical equations and the previous GLOF events, with scarcely documented detailed characteristics (Shrestha et al., 2023). Employing such proxy parameters is reasonable for this study as our aim is to provide an overview of GLOF risk in Bhutan based on the downstream impact. The modelled GLOF

scenarios for each lake are directly comparable, enabling an assessment of the overall and relative levels of danger, and representing a moderate scenario. We recognize however, that this is just one set of scenarios. Due to time and computational constraints, it was not feasible to simulate all potential variations. Future studies focusing on the detailed impact of specific glacial lakes or on specific downstream communities must be grounded on site-specific scenarios informed by situational triggering factors and dam composition and geometry. The study should also consider the site-specific worst-case scenario, considering the future climatic conditions."

A simulation of one or few past GLOFs and comparison of actual with simulated peak discharges would help gaining confidence into the model and its ability to realistically model GLOF dynamics. How can readers evaluate how well your model actually works? This would also enable to tune parameters and eventually study how sensitive the results are to uncertainties in the parameter values.

While evaluating model performance and parameter tuning based on the past event is necessary, it was not feasible due to the absence of previous GLOF events with observed characteristics in the region.

Based on our experience, we believe that reconstructed values from past events may not accurately represent future GLOFs, as each event is unique due to distinct geomorphological conditions and event-specific characteristics (Rinzin et al., 2025). Nevertheless, we are confident in the performance of the HEC-RAS model for simulating GLOF inundation, as evidenced by its successful application in a wide range of previous studies throughout the region (Maurer et al., 2020; Sattar et al., 2021; Rinzin et al., 2023; Sattar et al., 2022).

However, acknowledging the reviewer's concern, we will add sensitivity analysis considering the variability of Manning's n (between 0.040 and 0.070, the value range suggested by another reviewer) and lake size variation between 0.01 and 4.5 km². For lake size, we considered four scenarios: >0.05 km², 0.05 to >0.1 km², 0.1 to 1 km², and >1 km² in line with the size category we considered for calculating drainage volume. This sensitivity analysis will be conducted within the same basin to uniformly assess downstream impacts. Results of the sensitivity analysis will be added, and their implications will be discussed in the revised manuscript.

There are numerous instances were ambiguous or imprecise terms are used. Generally, I think that the terms threat and danger(ous) should be avoided, and that rather terms like hazard (probability of a potentially adverse event happening), exposure (how much are people or infrastructure within reach of a hazardous event), vulnerability (how susceptible are the exposed people or elements) and risk (the combination of the previous, quantitative metrics) should be used as they have a precise and measurable meaning. Threat and danger in turn have a qualitative and subjective meaning. Your work mainly aims to address the exposure of various elements at risk, and you quantify and aggregate the exposure so that it becomes an attribute of each lake. So, to this end, you quantify a lake-specific exposure index.

The same concern has also been raised by other reviewers. We have changed the terminologies consistently to the hazard, vulnerability, exposure and risk in all appearances in the revised manuscript.

**Minor comments:**

41: You state the number of 6907 fatalities, and backpedal later that this number is 80% attributed to a compound event involving the Chorabari outburst. The number of fatalities that can be clearly attributed to the Kedarnath event is probably very uncertain and much lower than those 80%. I would try to tone this more carefully, avoiding reporting numbers with high precision, that actually have a high uncertainty.

We have now amended this as:

"However, this reported number is highly uncertain, firstly due to scarce documentation of the past events, while 80% these reported deaths in HMA are associated with a single compounding event involving Chorabari glacial lake and widespread cloud outburst-induced debris flow within the basin in 2013"

49: Provide a definition of danger, in particular if your aim is to quantify it. Rather, as pointed out above, avoid this term entirely.

We replaced this 'danger' with 'risk'

81: This should be 31%, not 0.31%.

**Thanks. Amended.**

280: How was the HEC-RAS interfaced with? It would be great if you could add a technical description in a paragraph that details how you interfaced with HEC-RAS. I assume that you used the HEC-RAS controller to automate the tasks.

We did not use the HEC-RAS controller as we had to define separate upstream boundary conditions for each lake, which we believe is not straightforward, even using the HEC-RAS controller. Therefore, all models were set up manually, and simulations were done manually using computers in the geospatial laboratory at Newcastle University.

318f: Is it common to take the product of depth and velocity as damage level? Is it useful that the damage level of a water depth of 1 m and velocity of 5 m/s is the same for a water depth of 5 m and a velocity of 1 m/s?

It is not common, but it is a robust approach to account for both depth and velocity to calculate the damage level, especially in mountainous terrain conditions, since flow velocity plays a critical role in causing damage to infrastructure. Regarding the discrepancy where the impact of a water depth of 1 m and a velocity of 5 m/s is considered the same as that of a water depth of 5 m and a velocity of 1 m/s, we can say that the damage level might also depend on how long the structure remains submerged in water or kinetic energy associated with the debris content in the flow. However, for this first-order risk assessment, we believe it is reasonable to assume the damage level is the same, since we do not account for other details such as the content and actual value of the building.

540: It would be helpful to use a stringent and precise terminology here. What is devastating in comparison to damaging? Was the Missoula flood devastating, but not damaging, because no humans were affected (not sure whether this is true)? In simple terms, the risk of GLOFs is mainly determined by the exposed elements at risk, not by their hazard?

594f: I don't think that your approach challenges traditional susceptibility analyses. Rather, your approach may complement them. In contrast to susceptibility studies, your analysis assumes that the outburst probability is homogeneous, thus neglecting any variations in dam stability and lake exposure to avalanches and landslides.

Thanks for this suggestion. We have now corrected this statement as follows:

"Our study complements the conventional dangerous glacial lake assessments approach by redefining which glacial lakes pose the greatest danger to the downstream settlements"

719-721: Considering a risk framework, this is a somewhat trivial statement.

This is now corrected as follows:

However, the destruction and damage caused during the GLOF events are not only due to the hazard but also depend on their interaction with downstream exposed elements.

729: Please avoid the high precision of numbers when their estimates are prone to large uncertainties.

Thanks. It is now corrected as:

The analysis revealed that over 20,000 people, 2600 buildings, as well as other infrastructure such as roads and bridges and farmland are exposed to GLOF in Bhutan

Table S1: As the table spans several pages, it would be great to have the header row of this table on each page.

Thanks, amended as suggested.

Table S2: Be consistent in the number of digits that you report. Up to 8 digits behind the decimal point suggest an accuracy that you probably don't have in your measurements. When reporting counts (Bridges), use integers.

Thanks, the table is amended as suggested in the revised manuscript.